# A method for an unbiased estimate of cross-ancestry genetic correlation using individual-level data

Md. Moksedul Momin [1,2,3,4], Jisu Shin [1,2,5,6], Soohyun Lee[7], Buu Truong [1], Beben Benyamin [1,2,4] & S. Hong Lee [1,2,4] ✉

Cross-ancestry genetic correlation is an important parameter to understand the genetic relationship between two ancestry groups. However, existing methods cannot properly account for ancestry-specific genetic architecture, which is diverse across ancestries, producing biased estimates of cross-ancestry genetic correlation. Here, we present a method to construct a genomic relationship matrix (GRM) that can correctly account for the relationship between ancestry-specific allele frequencies and ancestry-specific allelic effects. Through comprehensive simulations, we show that the proposed method outperforms existing methods in the estimations of SNP-based heritability and cross-ancestry genetic correlation. The proposed method is further applied to anthropometric and other complex traits from the UK Biobank data across ancestry groups. For obesity, the estimated genetic correlation between African and European ancestry cohorts is significantly different from unity, suggesting that obesity is genetically heterogenous between these two ancestries.

Complex traits are polygenic and influenced by environmental factors[1,2], which can be distinguished from Mendelian traits that are regulated by single or few major genes and minimal environmental influences. In humans, causal loci and their effects on complex traits are not uniformly distributed across populations such that the same trait can be genetically heterogenous between two ancestry groups[3–5]. Studies of cross-ancestry genetic correlation are therefore important to understand the genetic relationship between two ancestry groups when investigating a complex trait[2,3]. Understanding genetic correlation between two ancestry groups can inform us on how well we can predict a complex disease in one ancestry group based on the information on another.

Genome wide association studies (GWAS) have been successful in identifying variants associated with causal loci, and can also provide a useful resource to investigate cross-ancestry genetic correlations. Using GWAS datasets, genomic relationships between samples across multiple ancestry groups can be constructed based on genome-wide single nucleotide polymorphisms (SNPs), which can be fitted with phenotypes across ancestries in a statistical model[3,6]. This paradigm-shift approach to estimate cross-ancestry genetic correlations have greatly increased the potential to dissect the latent genetic relationship between ancestry groups for complex traits.

Most GWAS have focused on European ancestry samples (>80%)[7–9] although Europeans represent only 16% of the global

[1]Australian Centre for Precision Health, University of South Australia, Adelaide, SA 5000, Australia. [2]UniSA Allied Health and Human Performance, University of South Australia, Adelaide, SA 5000, Australia. [3]Department of Genetics and Animal Breeding, Faculty of Veterinary Medicine, Chattogram Veterinary and Animal Sciences University (CVASU), Khulshi, Chattogram 4225, Bangladesh. [4]South Australian Health and Medical Research Institute (SAHMRI), University of South Australia, Adelaide, SA 5000, Australia. [5]Center for Public Health Genomics, University of Virginia, Charlottesville, VA 22908, USA. [6]Department of Biochemistry and Molecular Genetics, University of Virginia, Charlottesville, VA, USA. [7]Division of Animal Breeding and Genetics, National Institute of Animal Science (NIAS), Cheonan, South Korea. ✉e-mail: Hong.Lee@unisa.edu.au

population[10–12]. Because of large samples, the estimated SNP associations in Europeans are far more accurate than those in other ancestries. As a matter of fact, the performance of polygenic risk prediction depends on the accuracy of estimated SNP associations, minor allele frequencies (MAF), linkage disequilibrium (LD)[13,14], and environmental heterogeneity[15], causing ancestral disparity in genetic prediction accuracy[7]. Therefore, cross-ancestry genetic studies are urgently required to bridge the disparity, e.g., estimated cross-ancestry genetic correlations may be able to inform if SNP effects estimated in European ancestry samples can be useful to predict the phenotypes of samples in other ancestry groups.

To date, several studies have been undertaken to estimate cross-ancestry genetic correlations between diverse ancestry groups for a range of complex traits[2,3,16–18]. For example, Yang et al.[6] estimated the cross-ancestry genetic correlation between East Asians and Europeans for ADHD, where ancestry-specific allele frequencies were used to standardise samples' genotype coefficients in estimating their genomic relationships. This method of cross-ancestry genomic relationships has been widely used for cross-ancestry genetic studies[3,19]. However, the method cannot account for the trait genetic architecture specific to each ancestry group, which is a function of the relationship between the genetic variance and allele frequency (one important aspect of a heritability model[20,21]). With an incorrect heritability model, estimated genetic variances within and covariance between ancestry groups are biased[20,21], and hence the cross-ancestry genetic correlations cannot be correctly estimated. Another method based on GWAS summary statistics has been introduced (Popcorn)[2]. However, this method also cannot correctly account for diverse genetic architectures across ancestry groups.

Here, we develop a novel method that can properly account for ancestry-specific genetic architecture and ancestry-specific allele frequency in estimating a genomic relationship matrix (GRM). In addition, we revisit the existing theory to correct mean bias in genomic relationships. In simulations, the SNP-based heritability and cross-ancestry genetic correlation estimated from our proposed method are shown to be unbiased, whereas other existing methods can generate biased estimates. We apply the proposed method to six anthropometric and five other complex traits from the UK Biobank data, e.g., standing height, body mass index (BMI), waist-hip ratio, basal metabolic rate, body fat percentage, pulse rate and education. For each trait, we estimate SNP-based heritabilities and cross-ancestry genetic correlations across five groups, i.e., White British, Other Europeans, South Asian and African ancestries, and a mixed ancestry group.

## Results

### Overview

Our main aim is to unbiasedly estimate cross-ancestry genetic correlation for a complex trait, using common SNPs (e.g., those with MAF > 0.01) presented for both populations. We used publicly available data from the UK Biobank. Participants of the UK Biobank were

stratified into multiple ancestries (White British, Other European, South Asian, African, and mixed ancestry cohorts) according to their underlying genetic ancestry based on a principal component analysis (see Supplementary Fig. S1)[22]. In each ancestry group, we assume that the relationship between genetic variance and allele frequencies (heritability model) varies, i.e. the genetic variance at the $i^{th}$ genetic variant ($v_i$) can be expressed as

$$\mathrm{Var}(v_i) = 2p_i(1-p_i)\gamma_i^2 = (\beta_i)^2 \times [2p_i(1-p_i)]^{(1+2\alpha)} \qquad (1)$$

where $\gamma_i = \beta_i \times [2p_i(1-p_i)]^{\alpha}$[20,21,23] is the allele effect size ($\beta_i$) modified by a function of the reference allele frequency ($p_i$) and $\alpha$ that is the scale factor determining the genetic architecture of complex traits in each ancestry groups (Methods). Note that with $\alpha = 0$, Eq. (1) becomes the classical model of Falconer and Mackay (1996)[24]. By assuming that the genetic variance of causal variant is constant across minor allele frequencies (MAF) spectrum, a heritability model with $\alpha = -0.5$ has been widely used[25–27]. However, Speed et al.[20,21,23] reported a different $\alpha$ value, e.g., $\alpha = -0.125$ for anthropometric traits, using multiple European cohorts. While it is intuitive that $\alpha$ values can be varied across ancestry groups, it has not been well studied. Accounting for $\alpha$ correctly, we can disentangle $\beta_i$ from $\gamma_i$ so that we may be able to estimate the correlation of per-allele effect size in the original scale (see Methods).

First, we determine an optimal $\alpha$ value for each ancestry group, comparing model-fit (maximum log-likelihood) of various heritability models with different $\alpha$ values for the 6 anthropometric traits from the UK Biobank (see Methods). Second, we simulate phenotypes based on the UK Biobank genotypic data to assess if SNP-based heritability and cross-ancestry genetic correlation are unbiasedly estimated. In the simulation, various $\alpha$ values are used to generate allelic effects of SNPs in various ancestry groups, and the correlation of SNP effects between ancestry groups varies between 0 and 1 (see Methods). For simulated phenotypes of multiple ancestry groups, we estimate SNP-based heritability and cross-ancestry genetic correlation, using bivariate GREML[26–28] with four existing methods to construct GRM as shown in Table 1. The existing methods (GRM1 – 4 in Table 1) are individual-level data-based methods that are known to provide more accurate estimates, compared to summary statistics-based methods[21,29,30]. In addition to the existing methods, we use a novel method to estimate GRM in the cross-ancestry analysis in which we use ancestry-specific $\alpha$ value and ancestry-specific allele frequency, so that the estimation model matches with the ancestry-specific genetic architecture (see Methods). The equation for the proposed method can be written as

$$A_{ij} = \frac{1}{\sqrt{d_{k\_i} d_{k\_j}}} \sum_{l=1}^{L} (x_{il} - 2p_{lk\_i})(x_{jl} - 2p_{lk\_j}) var(x_{lk\_i})^{\alpha_{k\_j}} var(x_{lk\_j})^{\alpha_{k\_j}} + f_{bias_l}$$

$$(2)$$

where $x_{il}$ and $x_{jl}$ are SNP genotypes for the $i^{th}$ and $j^{th}$ individuals at the $l^{th}$ SNP, $p_{lk\_i}$ and $p_{lk\_j}$ are the allele frequencies at the $l^{th}$ SNP ($l = 1 - L$,

**Table 1 | Four existing methods to estimate cross-ancestry genetic correlation, compared to the proposed method**

| Methods | Scale factor ($\alpha$) | SNPs | Allele frequency | Reference | Equation No. |
|---|---|---|---|---|---|
| GRM1 | −0.5 | All[a] | overall-average[b] | [3,25,26] | (5, 7) |
| GRM2 | −0.5 | common only[c] | overall average | [3,25,26] | (5,7) |
| GRM3 | −0.5 | All | ancestry-specific[d] | [3,6] | (10) |
| GRM4 | −0.5 | common only | ancestry-specific | [3,6] | (10) |
| Proposed method | ancestry-specific[e] | common only | ancestry-specific | | (2) |

[a]Using all SNPs from both ancestry groups.
[b]Allele frequencies estimated from the combined population of both ancestry groups when scaling the genotypes.
[c]Using only common SNPs presented for both ancestry groups after QC including ancestry specific MAF > 0.01.
[d]Allele frequencies estimated from each population when scaling the genotypes.
[e]Different $\alpha$ value specific to each ancestry group can be used when scaling the genotypes.

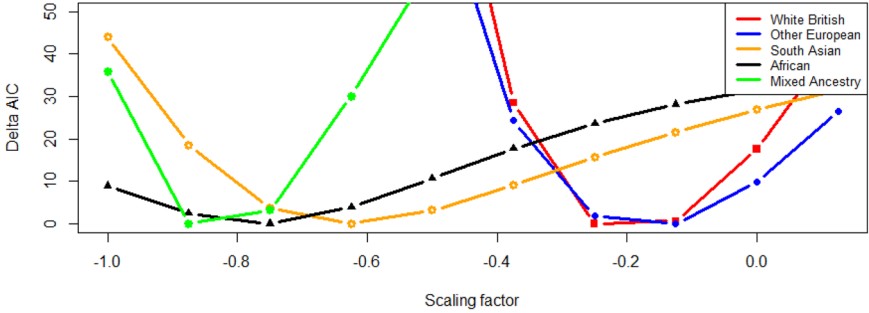

**Fig. 1 | Determining optimal scale factors ($\alpha$) for 5 different ancestry groups using GCTA-$\alpha$ model.** GCTA-$\alpha$ model assumes that all SNPs have an equal contribution to the heritability estimation and $\alpha$ value varies across ancestries[20,21]. $\Delta$AIC values from GCTA-$\alpha$ models are plotted against scaling factors, $\alpha$, for each ancestry group. The lowest AIC (i.e., $\Delta$AIC = 0) indicates the best model. The sample sizes are 30,000, 26,457, 6,199, 6,179 and 11,797 for White British, Other European, South Asian, African, and mixed ancestry groups, respectively.

where L is the number of SNPs) estimated in the two ancestry groups, $k\_i$ and $k\_j$, to which the $i^{th}$ and $j^{th}$ individuals belongs, and $\alpha_{k\_i}$ and $\alpha_{k\_j}$ are the scale factors for the two ancestry groups, $x_{lk\_i}$ and $x_{lk\_j}$ are all individual genotypes at the $l^{th}$ SNP in the two ancestry groups, $d_{k\_i}$ is the expectation of the diagonals, $d_{k\_i} = L\{\mathbb{E}[(x_i - 2p_{lk\_i})^2 var(x_{lk\_i})^{2\alpha}] + \frac{1}{n_{k\_i}} var(x_{lk\_i})^{(1+2\alpha_{k\_i})}\}$, and $f_{bias_l}$ is the bias factor at the $l$th SNP, $f_{bias_l} = \frac{1}{\sqrt{n_{k\_i}n_{k\_j}}} var(x_{lk\_i})^{(0.5+\alpha_{k\_i})} var(x_{lk\_j})^{(0.5+\alpha_{k\_j})}$ where $n_{k\_i}$ and $n_{k\_j}$ are the number of individuals in the two ancestry groups. The term, $f_{bias_l}$, can correct for the mean bias in the existing equations[25,26] (see Methods and Supplementary Table 1). In addition, we note that using var($x_l$), instead of its expectation ($2p_l(1 - p_l)$), is more robust[31] (Supplementary Table 2). Equation 2 can account for heterogenous $\alpha$ across ancestries, so that per allele effect size can be estimated accurately in the original scale in each ancestry, therefore, the correlation of per allele effect size is unbiasedly estimated, i.e., disentangling $\beta_i$ from $\gamma_i$.

However, existing methods estimate heritability and genetic correlation based on $\gamma_i$. More details for the derivation of Eq. 2 are in Methods. It is noted that if there is no varying alpha between ancestries and alpha is fixed as −0.5, the proposed method is equivalent to GRM4. Finally, we analyse real data using the proposed method (Eq. 2) to estimate SNP-based heritability and cross-ancestry genetic correlation for 6 anthropometric traits across different ancestries using bivariate GREML.

## Determination of scale factor ($\alpha$) across ancestries

We compared the Akaike information criteria (AIC) values of heritability models with varying $\alpha$ values to determine which $\alpha$ value provides the best model fit (see Methods), which is analogue to the approach of Speed et al. (2017)[21]. In terms of LD weights, we contrasted two kinds of heritability models, i.e. GCTA-$\alpha$ vs. LDAK-thin-$\alpha$ model. GCTA-$\alpha$ model has no LD weights, whereas LDAK-thin-$\alpha$ model explicitly considers LD among SNPs.

When using GCTA-$\alpha$ model, we observed that AIC values with $\alpha = -0.25$, −0.125, −0.625, −0.75 and −0.825 were lowest for White British, Other European, South Asian, African, and mixed ancestry cohorts, respectively (Fig. 1 and Supplementary Tables 3–7). When considering LDAK-thin-$\alpha$ model, we estimated optimal $\alpha$ values as −0.25, −0.125, −0.50, −0.625 and −0.75 for White British, Other European, South Asian, African, and mixed ancestry cohorts, respectively (Supplementary Fig. 2 and Supplementary Tables 3–7).

When comparing GCTA-$\alpha$ and LDAK-thin-$\alpha$ models, the AIC value of GCTA-$\alpha$ model was much smaller than that of LDAK-thin-$\alpha$ model for White British or Other European ancestry cohort (Supplementary Table 8). For South Asian ancestry cohort, the AIC of GCTA-$\alpha$ model was slightly lower than LDAK-thin-$\alpha$ model when using the best $\alpha$ value = −0.625. In contrast, the AIC of

LDAK-thin-$\alpha$ model was generally lower than that of GCTA-$\alpha$ model for African or mixed ancestry cohort.

## Method validation by simulation

We simulated phenotypes based on the real genotypic data of multiple ancestry groups where the estimated $\alpha$ value for each ancestry group was used to generate SNP effects that were correlated between ancestry groups (Methods). In this simulation, we do not consider associations between the allelic effects and LD structure for any SNP, i.e., LDAK simulation model, because LDAK model was not particularly plausible for the genetic architecture of the traits, especially for White British, Other European and South Asian ancestry cohorts (Supplementary Table 8) and LDAK simulation model was not feasible for a bivariate framework. When using simulation models with $\alpha = -0.5$ for all ancestry groups, estimated SNP-based heritabilities were mostly unbiased for all the methods, GRM1-4 (Supplementary Table 9). Estimated genetic correlations from GRM3 and 4 were unbiased for all scenarios (Fig. 2 and Supplementary Table 9). However, estimated genetic correlations from GRM1 and 2 were biased when the true genetic correlation was high (Fig. 2 and Supplementary Table 9). This shows that estimated cross-ancestry genetic correlation can be biased unless ancestry-specific allele frequencies were properly accounted for.

To mimic real data, we simulated phenotypes based on the real genotypes of multiple ancestry groups, using realistic $\alpha$ values ($\alpha = -0.25$ for White British ancestry cohorts, $\alpha = -0.625$ for South Asian ancestry cohorts and $\alpha = -0.75$ for African ancestry cohorts) instead of using a constant $\alpha$ value across ancestries (Methods). For these simulated phenotypes, estimated cross-ancestry genetic correlations using four existing methods were biased when the true genetic correlation was 0.5 or higher between White British and African ancestry cohorts or between South Asian and African ancestry cohorts. Biased estimates were still observed even when ancestry-specific allele frequencies were considered in GRM3 and 4 (Fig. 3b, c). As expected, estimated SNP-based heritability were mostly biased because of misspecified $\alpha$ values when estimating GRMs (Supplementary Fig. 3 and Supplementary Table 10).

For the same simulated phenotypes, we applied the proposed method that accounts for ancestry-specific allele frequency and ancestry-specific $\alpha$ values (Eq. 2) and found that it provided unbiased estimates for both SNP-based heritability and cross-ancestry genetic correlation (Fig. 3 and Supplementary Table 11). It was noted that the proposed method was robust to different numbers of causal SNPs (Supplementary Table 12). When the causal SNPs were not 100% common between ancestries (Supplementary Table 13), estimated cross-ancestry genetic correlations were biased, and the biasedness could be reduced by using the proposed method (Supplementary Fig. 4).

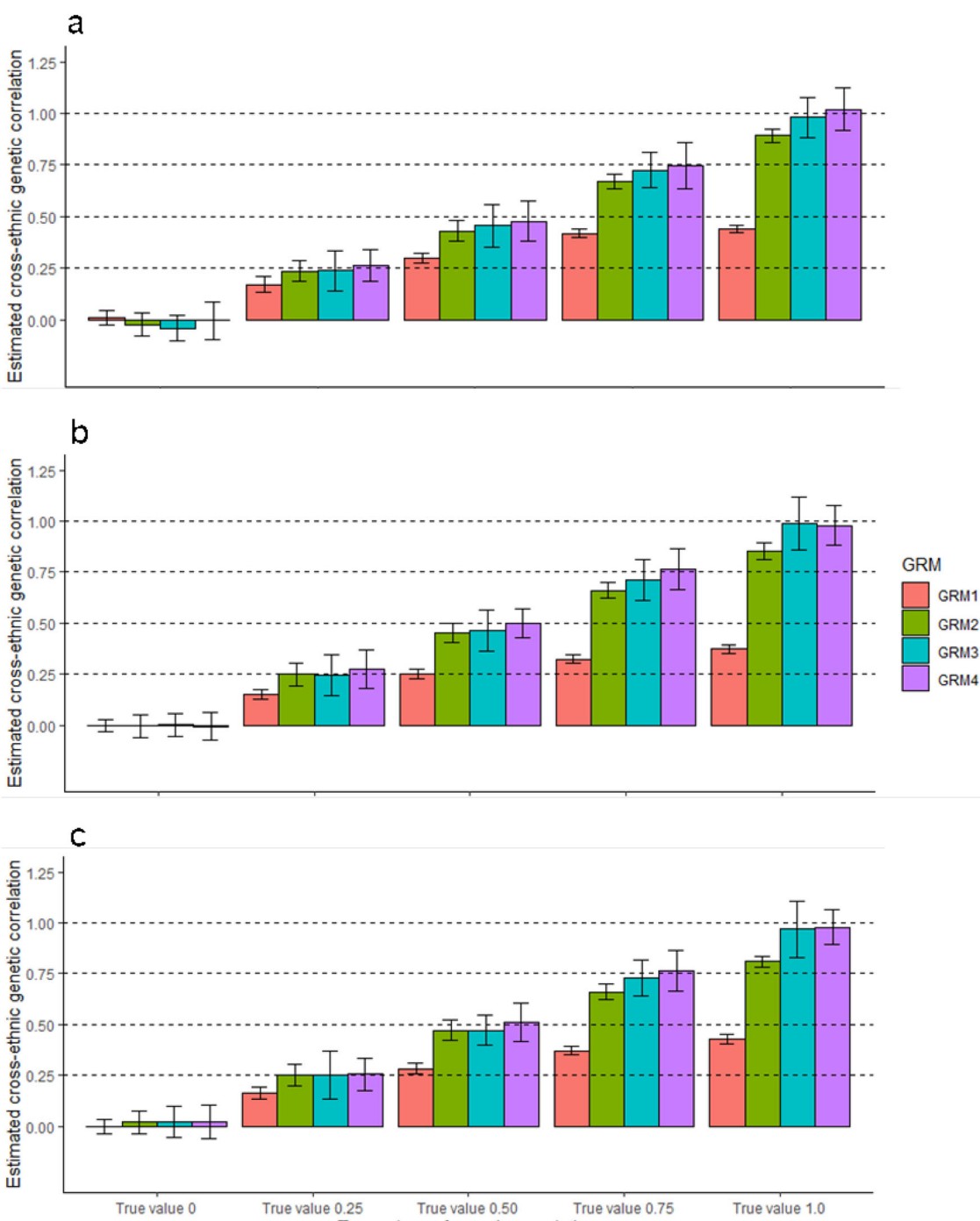

**Fig. 2 | Estimated cross-ancestry genetic correlations from four existing methods when α is fixed to −0.5 for both simulation and estimation.** The main bars represent the averages of estimated cross-ancestry genetic correlations, and the error bars indicate 95% confidence interval (CI) of the averages, which are derived from 500 experimental replicates that were independently carried out. In our simulations, we used three combinations for estimating cross-ancestry genetic correlation, i.e., **a** White British vs. South Asian, **b** White British vs. African and **c** South Asian vs. African ancestry cohorts. In each ancestry group, we used 500,000 SNPs that were randomly selected from HapMap3 SNPs after QC. To simulate phenotypes, we selected a random set of 1000 SNPs as causal variants, which were presented for both ancestry groups. We used α = −0.5 when scaling the

allelic effects by ancestry-specific allele frequency in each ancestry group. Various values of genetic correlation were considered (0, 0.25, 0.50, 0.75 and 1.0). In the estimation, the four methods (GRM1 − 4) used α = −0.5 (standard scale factor in GRM estimation). GRM1: Based on all SNPs between ancestry groups (791,581, 812,332 and 777,894 SNPs for the **a**−**c** with averaged allele frequency between two ancestries. GRM2: Based on common SNPs between ancestry groups (208,419, 187,668 and 222,106 SNPs for **a**−**c** with averaged allele frequency between two ancestries. GRM3: Based on all SNP between ancestry groups with ancestry-specific allele frequency. GRM4: Based on common SNP between ancestry groups with ancestry-specific allele frequency (see Table 1 for more details).

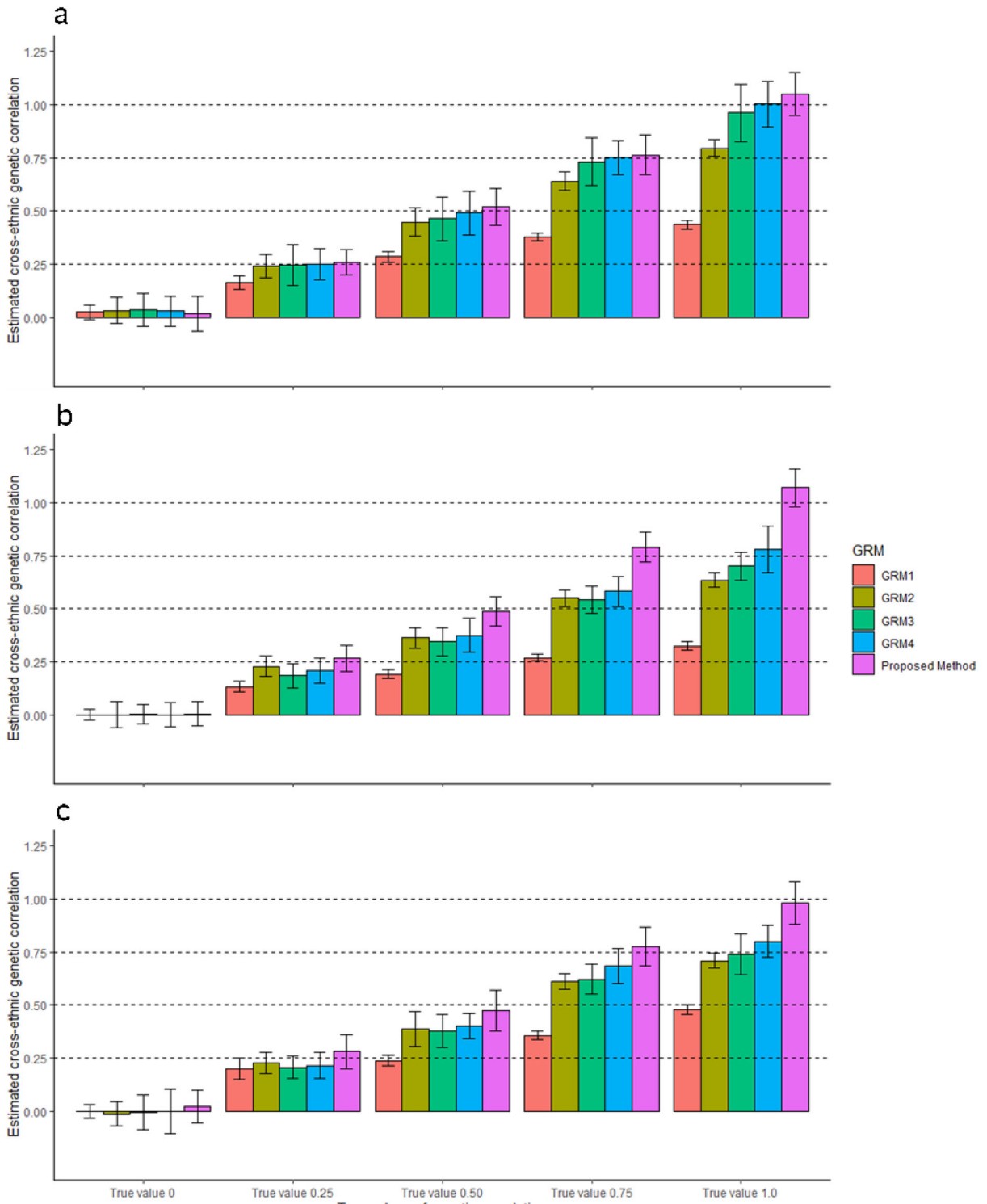

**Fig. 3 | Estimated cross-ancestry genetic correlations from four existing methods when varying $\alpha$ values across ancestry groups.** The main bars represent the averages of estimated cross-ancestry genetic correlations, and the error bars indicate 95% confidence interval (CI) of the averages, which are derived from 500 experimental replicates that were independently carried out. In our simulation, we used three combinations for estimating cross-ancestry genetic correlation, i.e., **a** White British vs. South Asian, **b** White British vs. African and **c** South Asian vs. African ancestry cohorts. In each ancestry group, we used 500,000 SNPs that were randomly selected from HapMap3 SNPs after QC. To simulate phenotypes, we selected a random set of 1000 SNPs as causal variants, which were presented for both ancestry groups. We used various $\alpha$ values that were specific to ancestries ($\alpha = -0.25$, $-0.625$ and $-0.75$ for White British, South Asian and African ancestry

cohorts, respectively) when scaling the allelic effects by ancestry-specific allele frequency in each ancestry group. Various values of genetic correlation were considered (0, 0.25, 0.50, 0.75 and 1.0). In the estimation, we used existing methods (GRM1 – 4) that used the standard scale factor $\alpha = -0.5$ in GRM estimation. GRM1: Based on all SNPs between ancestry groups (791,581, 812,332 and 777,894 SNPs for the **a**–**c** with averaged allele frequency between two ancestries. GRM2: Based on common SNPs between ancestry groups (208,419, 187,668 and 222,106 SNPs for **a**–**c** with averaged allele frequency between two ancestries. GRM3: Based on all SNP between ancestry groups with ancestry-specific allele frequency. GRM4: Based on common SNP between ancestry groups with ancestry-specific allele frequency (see Table 1 for more details). In addition, we applied the proposed method that used ancestry-specific $\alpha$ value and ancestry-specific allele frequency in GRM estimation.

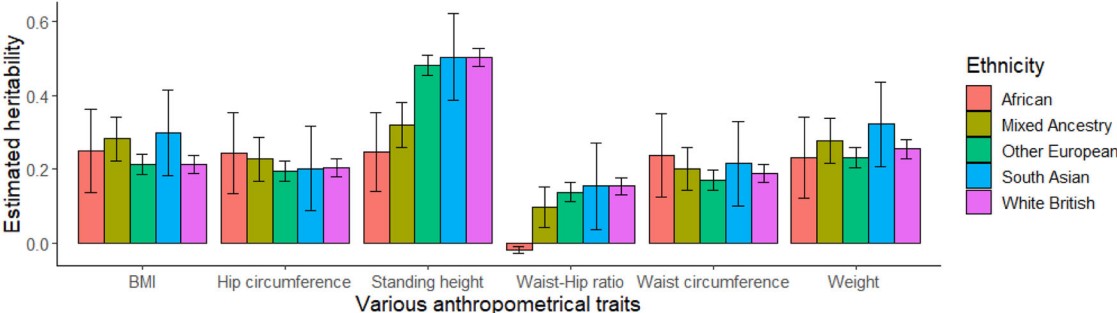

**Fig. 4 | Estimated SNP-based heritability for different anthropometric traits across ancestries.** The main bars indicate estimated SNP-based heritabilities, and the error bars indicate 95% confidence intervals (CI) that were derived from a linear mixed model. The total sample sizes used in the estimations were 30,000, 26,457, 6199, 6179, and 11,797 for White British, Other European, South Asian, African, and mixed ancestry cohorts.

We also compared Popcorn[2] and XPASS[32], which are GWAS summary statistics-based methods for cross-ancestry genetic analysis (Supplementary Notes), and found that they also generated biased estimates of cross-ancestry genetic correlation when using realistic $\alpha$ values in the simulation (Supplementary Table 14) with larger sample size (10,000 individuals). As expected, the computational efficiency of Popcorn is much higher than the proposed method and XPASS (Supplementary Table 15). It is noted that the computational efficiency of the proposed method can be increased further with parallel computing (Supplementary Table 16).

### SNP-based Heritability (h²) estimates for anthropometric traits using real data

The estimated SNP-based heritability for each of the anthropometric traits was presented in Fig. 4. The estimated SNP-based heritability of standing height was found to be highest in White British ancestry cohort (0.502, SE = 0.012) and lowest in African ancestry cohort (0.246, SE = 0.053). Heritability estimation in African ancestry cohort was significantly lower than White British ancestry cohorts ($p$ = 2.47e-06), Other European ancestry cohorts ($p$ = 2.30e-05) and South Asian ancestry cohorts ($p$ = 1.14e-03). The estimates for BMI and waist and hip circumference were generally high in African ancestry cohorts and low in Other European ancestry cohorts (Fig. 4). The estimated SNP-based heritability ranged from 0.231 (Other European ancestry cohorts) to 0.322 (South Asian ancestry cohorts) for weight, and from −0.02 (African ancestry cohorts) to 0.153 (South Asian ancestry cohorts) for waist-hip ratio. In contrast to height, there was no significant difference among ancestry-specific heritability estimates for BMI, waist circumference, hip circumference, waist hip ratio or weight.

### Cross-ancestry genetic correlations for anthropometric traits

Estimated cross-ancestry genetic correlations ($r_g$) between ancestry groups for 6 anthropometric traits are shown in Fig. 5. For BMI, the estimated genetic correlations between African and White British ancestry cohorts ($r_g$ = 0.672; SE = 0.131; $p$ = 1.22e-02) and between African and Other European ancestry cohorts ($r_g$ = 0.549; SE = 0.134; $p$ = 7.63e-04) were significantly different from 1 (Fig. 5a and Supplementary Table 17). This indicated that BMI is a genetically heterogenous between African and European ancestry cohorts. Estimated genetic correlations between African and South Asian or European and South Asian ancestry cohorts were low, but not significantly different from 1 (i.e., no evidence of genetic heterogeneity). As expected, the estimated genetic correlation between White British and Other European was not significantly different from 1 ($r_g$ = 1.081, SE = 0.043, $p$ = 5.96e-02).

For height, we observed a significant genetic heterogeneity between South Asian and African ancestry cohorts ($r_g$ = 0.356; SE = 0.169; $p$ = 1.38e-04), between Other European and South Asian ancestry cohorts ($r_g$ = 0.847; SE = 0.062; $p$ = 1.35e-02) and between

African and mixed ancestry cohorts ($r_g$ = 0.512; SE = 0.158; $p$-value = 2.01e-03) (Fig. 5b and Supplementary Table 18). Although estimated genetic correlations were lower than 1, there were no significant evidence for genetic heterogeneity between White British and South Asian ancestry cohorts ($r_g$ = 0.904; SE = 0.063; $p$ = 1.27e-01), and between White British and African ancestry cohorts ($r_g$ = 0.876; SE = 0.118; $p$ = 2.93e-01). White British and Other European ancestry cohorts were observed to be genetically homogeneous for the trait ($r_g$ = 1.01; SE = 0.018; $p$ = 5.78e-01).

Estimated cross-ancestry genetic correlations for waist circumference and hip circumference showed a similar pattern with BMI, i.e., there was significant evidence for genetic heterogeneity between White British and African ancestry cohorts and between Other Europeans and African ancestry cohorts (Fig. 5c, d; Supplementary Tables 19 and 20). The estimated genetic correlation between African and mixed ancestry cohorts was low, but not significantly different from 1. As the same as in BMI and height, there was no genetic heterogeneity between White British and Other European ancestry cohorts for both waist circumference and hip circumference.

We did not observe any significant heterogeneity across ancestries (genetic correlation estimate was not significantly different from 1) for waist-hip ratio (Fig. 5e and Supplementary Table 21). Non-estimable cross-ancestry genetic correlation was observed when using African ancestry cohort (NA in Fig. 5e) for which SNP-based heritability estimate was not significantly different from zero (Fig. 4 and Supplementary Table 19).

For weight, the estimated genetic correlation between African and Other European ancestry cohorts was significantly different from 1 ($r_g$ = 0.624; SE = 0.139; $p$ = 6.83e-03), indicating a significant genetic heterogeneity between these two ancestry groups (Fig. 5f and Supplementary Table 22). Although the estimations are not significant, we have estimated lower genetic correlations (far from 1) between White British and African ancestry cohorts, between African and mixed ancestry cohorts and between White British and South Asian ancestry cohorts.

### Application to a broad range of complex traits

We applied the proposed method to a broad range of phenotypic categories such as body fat, metabolic, cardiac and social behaviour traits, i.e., body fat percentage, whole body fat-free mass, basal metabolic rate, pulse rate and educational attainment. To estimate SNP-based heritabilities and cross-ancestry genetic correlations for these traits, we used trait- and ancestry-specific $\alpha$ that were obtained from the model comparisons[21,23] (Supplementary Tables 23–26 and Supplementary Fig. 11). The estimated SNP-based heritability is not significantly different across ancestries for all traits except that it is significantly higher in South Asian than Other European ancestry ($p$ = 6.07e-03) for educational attainment (Supplementary Fig. 12).

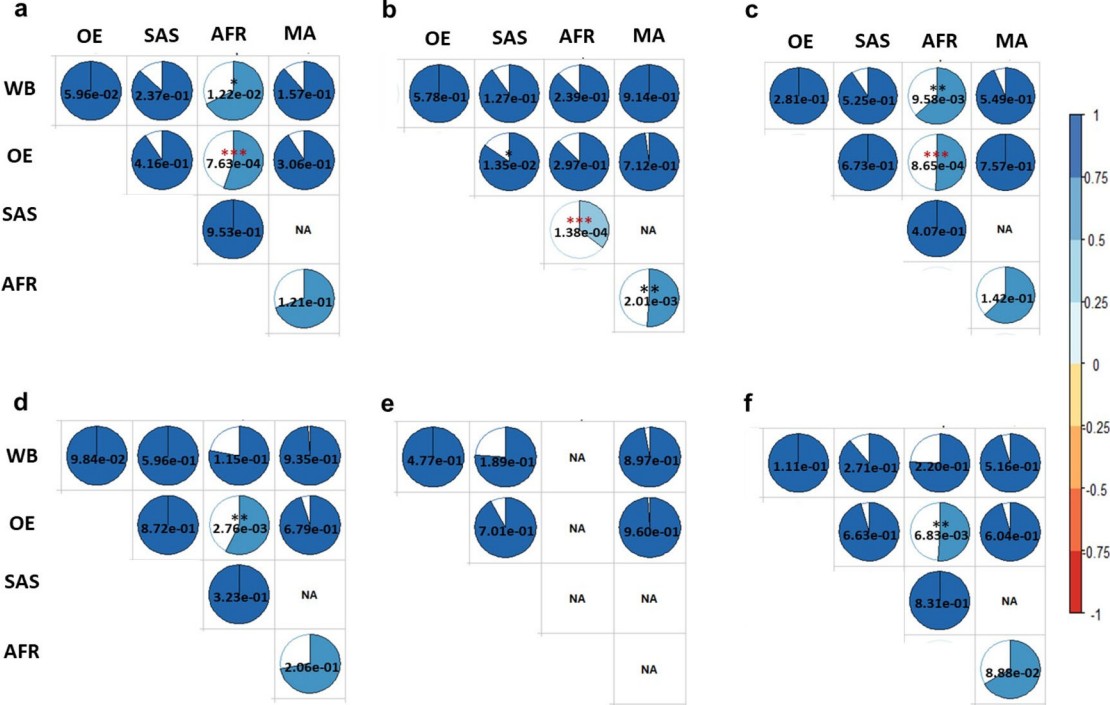

**Fig. 5 | Estimated cross-ancestry genetic correlations for anthropometric traits.** The phenotypes are **a** BMI **b** Standing height, **c** Waist circumference, **d** Hip circumference, **e** Waist-hip ratio and **f** Weight. The colour and size of each pie chart indicates the magnitude of estimated cross-ancestry genetic correlations. WB, OE, SAS, and AFR indicate White British, Other European, South Asian, and African ancestry cohorts. The value in each pie chart is a $p$-value (*, ** and *** indicate $p$ value <0.05, <0.01 and <0.001, respectively) based on Wald's test statistics for testing the null hypothesis of $r_g = 1$ (i.e., a two-sided test). Coloured asterisk indicates significantly different from 1 after Bonferroni correction (0.05/80). Non-estimable parameter is shown as NA, which was due to one ancestry group is nested within the other ancestry groups or estimated SNP-$h^2$ is zero for one ancestry group.

The pattern of estimated cross-ancestry genetic correlations is similar for basal metabolic rate and whole-body fat-free mass (Fig. 6 and Supplementary Tables 27 and 28). The estimated genetic correlation between Other European and African ancestries is significantly different from 1, indicating a significant genetic heterogeneity between the two ancestries for the trait ($r_g = 0.449$; SE = 0.109; $p = 4.30$e-07 for basal metabolic rate and ($r_g = 0.469$; SE = 0.107; $p = 6.95$e-07 for whole-body fat-free mass). In the case of body fat percentage, there is no significant genetic heterogeneity among ancestries although estimated cross-ancestry genetic correlations between African vs. White British, South Asian vs. White British, and South Asian vs. Other European are lower than 1 (Fig. 6 and Supplementary Table 29). We did not observe any significant genetic heterogeneity among the pairs of ancestry groups for pulse rate (Fig. 6 and Supplementary Table 30), noting that there might be a lack of power due to low SNP-based heritability estimates (Supplementary Fig. 12). For educational attainment, there is a highly significant genetic heterogeneity for all pairs of ancestries except the pair between White British and Other European (Fig. 6). Specifically, the estimated cross-ancestry genetic correlations are $r_g = 0.015$ (SE = 0.156; $p = 2.71$e-10) for African vs. White British, $r_g = -0.473$ (SE = 0.258; $p = 1.13$e-08) for African vs. South Asian, $r_g = 0.517$ (SE = 0.111; $p = 1.35$e-05) for White British vs. South Asian, $r_g = 0.495$ (SE = 0.124; $p = 4.64$e-05) for Other European vs. South Asian and $r_g = 0.262$ (SE = 0.182; $p = 5.01$e-05) for Other European vs. African ancestries (Fig. 6 and Supplementary Table 31).

## Discussion

We propose a novel method that provides unbiased estimates of ancestry-specific SNP-based heritability and cross-ancestry genetic correlations. This is possible because the proposed method correctly account for ancestry-specific genetic architectures or ancestry-specific heritability models. Our method provides a tool to dissect the ancestry-specific genetic architecture of a complex trait and can inform how genetic variance and covariance change across populations and ancestries. By using a meta-analysis across multiple ancestry groups[33] based on unbiased estimates of ancestry-specific heritability and cross-ancestry genetic correlations, we hope the current ancestry disparity and study bias in GWAS[7,10] can be reduced.

We investigated and found optimal $\alpha$ values for multiple ancestry groups, i.e. White British, Other Europeans, South Asian, African, and mixed ancestry cohorts, using six anthropometric traits from the UK Biobank. Interestingly, $\alpha$ values are distinct and not uniformly distributed across ancestries even for the same complex trait, that is the relationship between the allelic effects and allele frequency of the causal variants varies across ancestries. Per-allele effect size are not uniformly distributed, depending on genetic and environmental background or any unknown ancestry-specific factors. For example, if there are epistatic or interaction effects, selections on multiple loci can vary allele frequency, depending on per-allele effect sizes of loci. Moreover, SNP effects may reflect the level of association with causal variants such that per-allele effect size can be linearly correlated with allele frequency. Given our observation, it is clear that the heritability model should properly account for such diverse genetic architectures.

It was observed that the GCTA-$\alpha$ model outperformed LDAK-thin-$\alpha$ model for a more homogeneous population, such as White British, Other Europeans or South Asian ancestry cohort (Supplementary Table 8). For a less homogenous population such as mixed ancestry cohort, the LDAK-thin-$\alpha$ model was better than the GCTA-$\alpha$ model, implying that the choice of GCTA-$\alpha$ or LDAK-thin-$\alpha$ model might depend on the homogeneity of the population. It is noted that we used HapMap3 SNPs that have already excluded many redundant variants. A further study may be required to assess the performance of GCTA-$\alpha$ and LDAK-thin-$\alpha$ models with 1KG SNPs and other ancestry grouping, which is beyond the scope of this study.

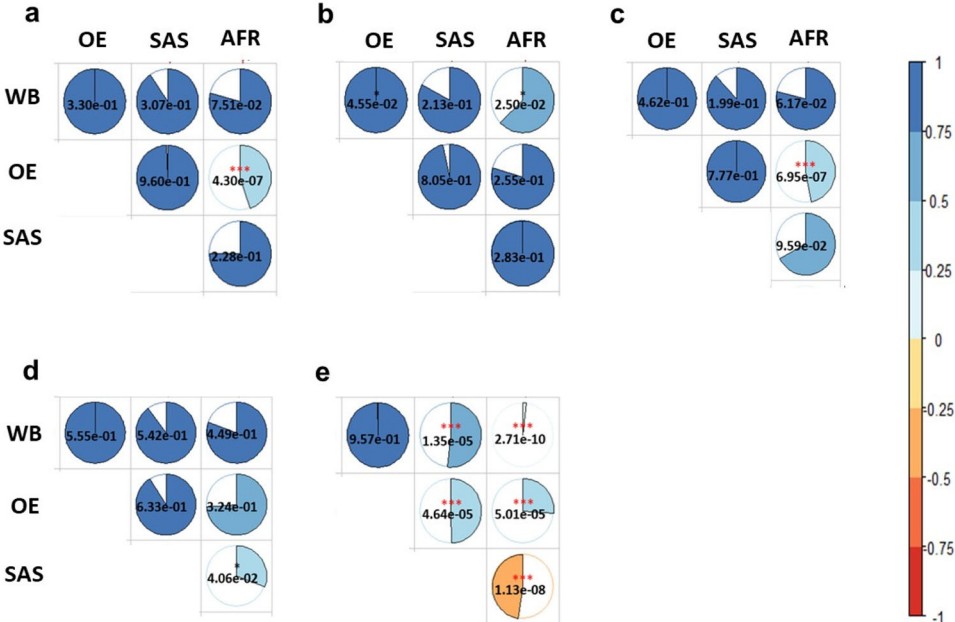

**Fig. 6 | Estimated cross-ancestry genetic correlations for a broader range of phenotypes.** The phenotypes are **a** Basal metabolic rate, **b** Body fat percentage, **c** Whole body fat-free mass, **d** Pulse rate and **e** Education. The colour and size of each pie chart indicates the magnitude of estimated cross-ancestry genetic correlations. WB, OE, SAS, and AFR indicate White British, Other European, South Asian, and African ancestry cohorts. The value in each pie chart is a $p$-value (*, ** and *** indicate $p < 0.05$, $<0.01$ and $<0.0001$, respectively) based on Wald's test statistics for testing the null hypothesis $r_g = 1$ (i.e., a two-sided test). Coloured asterisk indicates significantly different from 1 after Bonferroni correction (0.05/80).

The estimated SNP-based heritability of BMI in the White British ancestry cohort was not much different from previous studies[3,34], although our estimate was slightly lower probably because of using different $\alpha$ values. For the same reason, our estimate for African ancestry cohort was slightly higher than previous study[3]. For standing height, we observed that the estimated SNP-based heritability in African ancestry cohort was significantly lower than other ancestry groups, which agreed with previous studies[3,35]. This is probably due to the fact that African-specific causal variants are less tagged by the common SNPs or environmental effects are relatively large, compared to other ancestries, which requires further investigations. For other traits, our estimates were approximately agreed with a previous study[34,36,37] although using different $\alpha$ values.

Cross-ancestry genetic correlations can provide crucial information in cross-ancestry GWAS and cross-ancestry polygenic risk score prediction. We showed a significant genetic heterogeneity for obesity traits (BMI, weight and waist and hip circumferences) between African and European ancestry samples. For height, there was a significant genetic heterogeneity between African and South Asian ancestry samples. However, without considering ancestry-specific $\alpha$ values (i.e., GRM 4 from equation 5), the findings were changed and could be over-interpreted, e.g., there was additionally significant genetic heterogeneity between African and European ancestry cohorts (Supplementary Fig. 5), which agreed with Guo et al.[3] who used the same method as in GRM4 (equation 5).

LDSC based on GWAS summary statistics is computationally efficient and has been widely used in a single ancestry group (dominantly used in Europeans) in the estimation of SNP-based heritability and genetic correlations between complex traits[29]. Similar to LDSC, GWAS summary statistics-based cross-ancestry analyses (Popcorn[2] and XPASS[32]) has been used by several studies[16–18], to estimate cross-ancestry genetic correlations. However, as shown in the results from Brown et al.[2] and our simulation, the methods can be biased when the genetic architecture of a complex traits is diverse (i.e., when $\alpha$ values vary) across ancestries. S-LDXR[38] may be able to account for MAF-dependent genetic architecture, which, however, requires an arbitrary

stratification of SNPs into multiple MAF bins that was not considered in this study. It is also reported that Popcorn, MAMA and S-LDXR are likely to be suitable for a pair of European and East Asian ancestries, but they have not been explicitly tested when other ancestries are involved, e.g., African or South Asian ancestry.

There are several limitations in this study. Firstly, we used the GCTA-$\alpha$ model only in the real data analysis, assuming that all SNPs contributed equally to the heritability estimation[27,39]. We did not use the LDAK-thin-$\alpha$ model that required to prune SNPs within each ancestry group, which could substantially reduce the number of common SNPs between two ancestry groups in cross-ancestry genetic correlation analyses. Secondly, we did not consider MAF stratified, LDMS, or baseline model[40–42] in estimating cross-ancestry genetic correlations because it was developed to estimate SNP-based heritability, but not suitably designed to estimate genetic correlations. Furthermore, it is required to fit multiple random effects (i.e., multiple GRMs), which is not computationally efficient. Nevertheless, these advanced models can improve the estimations of ancestry-specific SNP-based heritability and cross-ancestry genetic correlation. Thirdly, we estimated optimal scale factors ($\alpha$) with a moderate sample size especially for South Asian or African ancestry cohorts, resulting in a relatively flat curve of ΔAIC values. A further study is required to estimate more reliable $\alpha$ for South Asian or African ancestry cohorts with a larger sample size. For educational attainment, non-genetic factors, such as socio-economic status and access to public education, might also contribute to the phenotypes. In this study, we did not consider how the genetic effects interact with such environmental factors, i.e., genotype-by-environment interaction, which may partly underlie the significant genetic heterogeneity across ancestries. A further study is required to elucidate how genotype-by-environment interaction cause cross-ancestry genetic heterogeneity, especially for educational attainment. Finally, when the causal SNPs may not be 100% common between ancestries, the estimated cross-ancestry genetic correlations should be carefully interpreted with caution although the estimates of the proposed method can be less biased, compared to existing methods.

In conclusion, we present a method to construct a GRM that can correctly account for the relationship between ancestry-specific allele frequencies and ancestry-specific allelic effects. As the result, our method can provide unbiased estimates of ancestry-specific SNP-based heritability and cross-ancestry genetic correlation. By applying our proposed method to anthropometric as well as other complex traits, we found that obesity is a genetically heterogenous trait for African and European ancestry cohorts, while human height is a genetically heterogenous trait between African and South Asian ancestry cohorts. For educational attainment, there is significant genetic heterogeneity between African and European, African and south Asian, and south Asian and European ancestries.

## Methods

### Statistical model

We use a bivariate linear mixed model (LMM) to estimate SNP-based heritability and cross-ancestry genetic correlation, using GWAS data comprising multiple ancestry groups. In the model, a vector of phenotypic observations for each ancestry group can be decomposed into fixed effects, random genetic effects and residuals. The LMM can be written as

$$\mathbf{y_i} = \mathbf{X_i b_i} + \mathbf{Z_i g_i} + \mathbf{e_i} \tag{3}$$

where $\mathbf{y_i}$ is the vector of phenotypic observations, $\mathbf{b_i}$ is the vector of fixed (environmental) effect with the incidence matrix $\mathbf{X_i}$, $\mathbf{g_i}$ is the vector of random additive genetic effects with the incidence matrix $\mathbf{Z_i}$ and $\mathbf{e_i}$ is the vector of residual effects for the $i^{th}$ ancestry group (i = 1 and 2).

The random effects, $\mathbf{g_i}$ and $\mathbf{e_i}$, are assumed to be normally distributed, i.e. $\mathbf{g_i} \sim N(0, \mathbf{A}\sigma_{g_i}^2)$ and $e_i \sim N(0, \mathbf{I}\sigma_{e_i}^2)$. The variance covariance matrix of observed phenotypes can be written as

$$\mathbf{V} = \begin{bmatrix} \mathbf{Z_1 A}\sigma_{g_1}^2 \mathbf{Z_1'} + \mathbf{I}\sigma_{e_1}^2 & \mathbf{Z_1 A}\sigma_{g_{12}}^2 \mathbf{Z_2'} \\ \mathbf{Z_2 A}\sigma_{g_{21}}^2 \mathbf{Z_1'} & \mathbf{Z_2 A}\sigma_{g_2}^2 \mathbf{Z_2'} + \mathbf{I}\sigma_{e_2}^2 \end{bmatrix} \tag{4}$$

where $\mathbf{A}$ is the genomic relationship matrix (GRM)[25,26,43], which can be estimated based on the genome-wide SNP information, and $\mathbf{I}$ is an identity matrix. The terms, $\sigma_{g_i}^2$ and $\sigma_{e_i}^2$, indicate the genetic and residual variance of the trait for the $i^{th}$ ancestry group, and $\sigma_{g_{12}}^2$ ($\sigma_{g_{21}}^2$) is the genetic covariances between the two ancestry groups. Note that it is not required to model residual correlation in $\mathbf{V}$ because there are no multiple phenotypic measures for any individual (no common residual effects shared between two ancestry groups).

### The variance of random additive genetic effects

Assuming that causal variants are in linkage equilibrium and that the phenotypic variance is $var(y) = 1$, the heritability can be written as

$$h^2 = \sigma_g^2 = \sum_{i=1}^{M} var(v_i)$$

where $var(v_i)$ is the genetic variance of the $i^{th}$ causal variant and M is the number of causal variant. The genetic variance at the $i^{th}$ genetic variant can be written as

$$var(v_i) = 2p_i(1 - p_i)\gamma_i^2$$

where $\gamma_i = \beta_i$ are allelic effects of the $i^{th}$ variant if we do not consider the relationship between $\beta_i$ and $p_i$, following Falconer and Mackay[24]. When considering the relationship between $\beta_i$ and $p_i$, $\gamma_i$ can be reparameterised as $\gamma_i = \beta_i \times [2p_i(1 - p_i)]^\alpha$ as suggested by previous studies[20,21,23]. This shows that, although $\beta_i$ is consistent, $\gamma_i$ can differ across ancestry groups that have different $p_i$ and/or $\alpha$. Therefore, as

shown in Eq. (1), the genetic variance at the $i^{th}$ genetic variant can be rewritten as

$$var(v_i) = 2p_i(1 - p_i)\gamma_i^2 = \beta_i^2 \times [2p_i(1 - p_i)]^{(1+2\alpha)}.$$

Assuming that the expectation of $\beta_i$ is $E(\beta_i) = 0$, the expectation of $var(v_i)$ can be expressed as

$$\mathbb{E}(var(v_i)) = \mathbb{E}(\beta_i^2) \times \mathbb{E}([2p_i(1 - p_i)]^{(1+2\alpha)}) = var(\beta_i) \times \mathbb{E}([2p_i(1 - p_i)]^{(1+2\alpha)})$$

where $var(\beta_i) = E(\beta_i^2) - E(\beta_i)E(\beta_i) = E(\beta_i^2)$.

This shows $\mathbb{E}(var(v_i)) = var(\beta_i)$ when using $\alpha = -0.5$ (i.e. the widely used assumption of constant variance across different MAF spectrum).

However, with various factors (selection, interaction, linkage disequilibrium, population stratification and so on), optimal $\alpha$ values vary across populations[21,23]. Therefore, although per-allele effect size ($\beta_i$) is constant, the actual effect, $\gamma_i = \beta_i \times (2p_i(1 - p_i)^\alpha)$, can be changed and may not be uniformly distributed across ancestries, depending on ancestry-specific factors. What we aim to estimate is $cor(\boldsymbol{\beta}_k, \boldsymbol{\beta}_l)$, the correlation between per-allele effect sizes for SNPs of the $k^{th}$ and $l^{th}$ ancestry groups, which is different from $cor(\boldsymbol{\gamma}_k, \boldsymbol{\gamma}_l)$.

### Genomic Relationship Matrix (GRM)

GRM is a kernel matrix and can be normalised with a popular form (VanRaden[25]; Yang et al.[26]) of

$$A_{ij} = \frac{1}{L} \sum_{l=1}^{L} \frac{(x_{il} - 2p_l)(x_{jl} - 2p_l)}{2p_l(1 - p_l)} \tag{5}$$

or an alternative form (VanRaden[25]) of

$$A_{ij} = \frac{\sum_{l=1}^{L}(x_{il} - 2p_l)(x_{jl} - 2p_l)}{\sum_{l=1}^{L} 2p_l(1 - p_l)} \tag{6}$$

where $A_{ij}$ is the genomic relationship between the $i^{th}$ and $j^{th}$ individuals, L is the total number of SNPs, $p_l$ is the reference allele frequency at the $l$th SNP, and $x_{il}$ is the genotype coefficient of the $i^{th}$ individual at the $l^{th}$ SNP.

Speed et al.[20,21] generalised these forms, introducing a scale factor that can determine the genetic architecture of a complex trait (aka heritability model). The generalised form is

$$A_{ij} = \frac{1}{d} \sum_{l=1}^{L} \left[(x_{il} - 2p_l)(x_{jl} - 2p_l)\right][2p_l(1 - p_l)]^{2\alpha} \tag{7}$$

where $\alpha$ is a scale factor, d is the expected diagonals, $d = L \cdot \mathbb{E}[(x_{il} - 2p_l)^2[2p_l(1 - p_l)]^{2\alpha}]$. When $\alpha = -0.5$, Eq. (7) is equivalent to Eq. (5), and when $\alpha = 0$, Eq. (7) becomes equivalent to Eq. (6). Note that each SNP can be weighted according to the LD structure (LDAK or LDAK-thin) if this weighting scheme better fits with the genetic architecture, which can be assessed by a model comparison[21].

However, these Eqs. (5, 6 and 7) do not account for correlation between $x_{il}$ and the estimated mean, i.e., $2p_l$, which can cause biased estimates of genomic relationships. With $\mu_l = 2p_l = \frac{\sum_{i=1}^{n} x_{il}}{n}$ and $\alpha = -0.5$, the diagonals can be expressed as

$$A_{ii} = \frac{1}{L} \sum_{l=1}^{L} \left[(x_{il} - \mu_l)[2p_l(1 - p_l)]^{-0.5}\right]^2 = \frac{1}{L} \sum_{l=1}^{L} [(x_{il} - \mu_l)^2 [2p_l(1 - p_l)]^{-1}]$$

where the expectation of the first term can be rewritten as

$$\mathbb{E}[(x_{il} - \mu_l)]^2 = \mathbb{E}([n \times x_{il} - x_{1l} - x_{2l} - \ldots - x_{nl}]^2/n^2)$$
$$= [n^2 \mathbb{E}(x_{il}^2) - 2n\mathbb{E}(x_{il}^2) + n\mathbb{E}(x_{il}^2) + n(n-1)\mathbb{E}(x_{il})\mathbb{E}(x_{il})$$
$$- 2n(n-1)\mathbb{E}(x_{il})\mathbb{E}(x_{il})]/n^2$$
$$= [n(n-1)\mathbb{E}(x_{il}^2) - n(n-1)[\mathbb{E}(x_{il})]^2])/n^2$$
$$= (1 - 1/n) \times var(x_{il}).$$

With $\mathbb{E}[var(x_{il})] = 2p_l(1 - p_l)$, the diagonals from Eqs. (5) or (7) can be expressed as

$$A_{ii} = 1 - \frac{1}{n},$$

which is deviated from 1 by a factor $1/n$. Without loss of generality, the biased factor with any $\alpha$ values can be written as

$$f_{bias} = -1/n \times var(x) \times [2p_l(1 - p_l)]^{2\alpha}.$$

In a similar manner, the off diagonals can be written as

$$A_{ij} = \frac{1}{L} \sum_{l=1}^{L} (x_{il} - \mu_l)(x_{jl} - \mu_l) * [2p_l(1 - p_l)]^{-1}$$

where the expectation of the first term can be rewritten as

$$\mathbb{E}[(x_{il} - \mu_l)(x_{jl} - \mu_l)]$$
$$= \mathbb{E}([n * x_{il} - x_{1l} - x_{2l} - , \ldots, x_{nl}][n \times x_{jl} - x_{1l} - x_{2l} - , \ldots, x_{nl}]/n^2)$$
$$= [n^2 \mathbb{E}(x_{il})\mathbb{E}(x_{jl}) - n\mathbb{E}(x_{il}^2) - n(n-1)\mathbb{E}(x_{il})\mathbb{E}(x_{jl})x_l]/n^2$$
$$= [n^2 \mathbb{E}(x_l)\mathbb{E}(x_l) - n\mathbb{E}(x_l^2) - n(n-1)\mathbb{E}(x_l)\mathbb{E}(x_l)]/n^2$$
$$= [n[\mathbb{E}(x_l)]^2 - n\mathbb{E}(x_l^2)]/n^2$$
$$= -1/n \times var(x)$$

With $\mathbb{E}[var(x_l)] = 2p_l(1 - p_l)$, the off diagonals from Eqs. (5) or (7) can be expressed as

$$A_{ij} = -\frac{1}{n},$$

which is deviated from 0 by a factor $1/n$. The biased factor with any $\alpha$ values is the same as in the diagonals, i.e. $f_{bias} = -1/n \times var(x) \times [2p_l(1 - p_l)]^{2\alpha}$.

Therefore, a revised formula, considering $f_{bias}$, should be

$$A_{ij} = \frac{1}{d} \left[ \sum_{l=1}^{L} \left[ (x_{il} - 2p_l)(x_{jl} - 2p_l) \right] [2p_l(1 - p_l)]^{2\alpha} + \frac{1}{n} var(x_l) [2p_l(1 - p_l)]^{2\alpha} \right] \tag{8}$$

where $d$ is redefined as $d = L\{\mathbb{E}[(x_i - 2p)^2 [2p(1-p)]^{2\alpha}] + \frac{1}{n} var(x_l) [2p_l(1 - p_l)]^{2\alpha}\}$. It is noted that with a sufficient sample size (>1000), the difference between Eqs. (7) and (8) is negligible.

Furthermore, $2p_l(1 - p_l)$ is the expectation of $var(x)$. Unless the samples are completely homogenous, the expectation is an approximation of $var(x)$. So, $var(x)$ should be used instead of the expectation $2p(1-p)$. Therefore, the formula can be rewritten as

$$A_{ij} = \frac{1}{d} \left[ \sum_{l=1}^{L} \left[ (x_{il} - 2p_l)(x_{jl} - 2p_l) \right] var(x_l)^{2\alpha} + \frac{1}{n} var(x_l)^{(1+2\alpha)} \right] \tag{9}$$

with $d = L\{\mathbb{E}[(x_i - 2p)^2 var(x_l)^{2\alpha}] + \frac{1}{n} var(x_l)^{(1+2\alpha)}\}$

### GRM for cross-ancestry genetic analyses

Yang et al.[6] proposed a GRM method that uses ancestry-specific allele frequency to be applied to cross-ancestry genetic analyses, which can

be expressed as

$$A_{ij} = \frac{1}{L} \sum_{l=1}^{L} \frac{(x_{il} - 2p_{lk_i})(x_{jl} - 2p_{lk_j})}{2\sqrt{p_{lk_i}(1 - p_{lk_i})p_{lk_j}(1 - p_{lk_j})}} \tag{10}$$

where $p_{lk_i}$ and $p_{lk_j}$ are the allele frequencies at the $l^{th}$ SNP estimated in the $k_i$ and $k_j$th ancestry groups to which the $i^{th}$ and $j^{th}$ individuals belongs. When estimating cross-ancestry genetic correlation for Attention Deficit Hyperactivity Disorder (ADHD) between European and East Asian (Han Chinese), Yang et al.[6] considered the standard scale factor for both ancestry groups ($\alpha = -0.5$ in White European and East Asian). Similarly, Guo et al.[3] also used Eq. (10) to estimate cross-ancestry genetic correlation between White British and African ancestry cohorts, assuming $\alpha$ value was constant across these ancestry groups ($\alpha = -0.5$).

It is intuitive that $\alpha$ value can be changed across ancestry groups because of genetic drift, natural selection, and various selection pressures. Combined with a revised formula derived above (Eq. (9)), a novel GRM equation for cross-ancestry genetic analyses, which accounts for ancestry-specific $\alpha$ and ancestry-specific allele frequency, can be proposed as Eq. (2),

$$A_{ij} = \frac{1}{\sqrt{d_{k\_i} d_{k\_j}}} \sum_{l=1}^{L} (x_{il} - 2p_{lk\_i})(x_{jl} - 2p_{lk\_j}) var(x_{lk\_i})^{\alpha_{k\_i}} var\left(x_{lk\_j}\right)^{\alpha_{k\_j}} + f_{bias_l}$$

This proposed method allows us to account for heterogenous $\alpha$ across ancestries, which can provide unbiased estimates of per allele effect size on the original scale in each ancestry, i.e. correctly estimating $\beta_i$ in Eq. (1). Therefore, the correlation of per allele effect size can be unbiasedly estimated, nothing again that we are interested in estimating $cor(\boldsymbol{\beta}_k, \boldsymbol{\beta}_l)$, correlation between per-allele effect sizes for SNPs of the $k^{th}$ and $l^{th}$ ancestry groups, not $cor(\boldsymbol{\gamma}_k, \boldsymbol{\gamma}_l)$.

### Data source and quality control

We tested this proposed method in real genotypic and phenotypic data obtained from the second release of the UK Biobank (https://www.ukbiobank.ac.uk/)[44]. The genotypic data were imputed based on Haplotype Reference Consortium reference panel[45]. The UK Biobank data comprised 488,377 participants and 92,693,895 SNPs. The participants were grouped into four ancestry groups (White British, Other European, South Asian and African ancestries) and one mixed ancestry cohort, according to their genetic ancestry estimated from a principal component analysis based on the genome-wide SNP infromation[22]. Mixed ancestry cohort includes individuals from South Asian ancestry, some of white and black African and white and black Caribbean ancestries and those individuals assigned as other ancestry groups in UK biobank (Supplementary Fig. 1). We grouped Other Europeans separately from White British because the former is more diverse than the latter[46]. (Supplementary Fig. 1). We did not include individuals who do not know their ancestry and who prefer not to answer (UK Biobank codes are −1 and −6). Gender mismatch and sex chromosome aneuploidy were also excluded during the quality control (QC) process.

We performed additional stringent QC in each of the ancestry groups. The QC criteria include an INFO score (an imputation reliability) ≥0.6[47–49], SNP missingness <0.05, minor allele frequency (MAF) > 0.01, Hardy–Weinberg equilibrium $p > 10^{-04}$. We also excluded individuals outside ± SD of the population mean for first and second ancestry principal components. Individuals with genetic relatedness ≥0.05 were excluded from each ancestry group using PLINK[50]. In the analysis, we retained HapMap3 SNPs only as these are high in quality and well calibrated to dissect genetic architecture of complex traits[51,52].

The initial sample size was 430,301, 29,023, 7449, 7647 and 16,615 for White British, Other European, South Asian, African, and mixed

ancestry cohorts, respectively. After quality control, the cleaned data includes 288,837, 26,457, 6199, 6179 and 11,797 participants, and the total number of SNP was 1,154,490, 1,148,504, 939,512, 729,534 and 513,362 for White British, Other European, South Asian, African, and mixed ancestry cohorts. For White British ancestry cohort, we randomly selected 30,000 from the QCed 288,837 individuals for estimating cross-ancestry genetic correlations because it is computationally feasible in multiple analyses paring with other ancestry groups.

### Phenotype simulation

The phenotypes of each ancestry group were simulated using a bivariate linear mixed model (Eq. 3, i.e. $y_i = X_i b_i + Z_i g_i + e_i$). In the simulation, 1000 individuals and 500,000 SNPs were randomly chosen for each ancestry group. To simulate phenotypes, we randomly selected 1000 SNPs that were common between the two ancestry groups and assigned allelic effects to them. According to Eq. (4), a multivariate normal distribution was used to draw allelic effects of the 1000 SNPs with mean $\begin{bmatrix} \bar{g}_1 \\ \bar{g}_2 \end{bmatrix} = \begin{bmatrix} 0 \\ 0 \end{bmatrix}$, and genetic covariance matrix as $\begin{bmatrix} \sigma_{g_1}^2 & \sigma_{g_{12}}^2 \\ \sigma_{g_{21}}^2 & \sigma_{g_2}^2 \end{bmatrix} = \begin{bmatrix} 0.5 & 0, 0.25, 0.50, 0.75 \text{ or } 1.0 \\ 0, 0.25, 0.50, 0.75 \text{ or } 1.0 & 0.5 \end{bmatrix}$.

According to Eq. (1), the allelic effects of the $i^{th}$ SNPs were scaled by actual variance $[var(x_i)]^\alpha$, where $p_i$ is the reference allele frequency, noting that $\alpha$ and $p_i$ can vary between the two ancestry groups. Individual genetic values (i.e. polygenic risk scores) are the sum of their genotype coefficients of the 1000 causal SNPs, weighted by the allelic effects. The simulated phenotypes were generated as the summation of the true genetic values and the residual effects (Eq. 4) which were obtained from a multivariate normal distribution with mean $\begin{bmatrix} \bar{e}_1 \\ \bar{e}_2 \end{bmatrix} = \begin{bmatrix} 0 \\ 0 \end{bmatrix}$ and the residual covariance matrix $\begin{bmatrix} \sigma_{e_1}^2 & \sigma_{e_{12}}^2 \\ \sigma_{e_{21}}^2 & \sigma_{e_2}^2 \end{bmatrix} = \begin{bmatrix} 0.5 & 0 \\ 0 & 0.5 \end{bmatrix}$. Hence, the true heritability was set as 0.5 for both ancestry groups in the simulation based on the bivariate linear mixed model.

To validate the proposed method of GRM estimation, we considered scenarios with the number of causal SNPs 100, 1000, 10,000 and 100,000. Phenotypic data were simulated based on standard $\alpha$ and estimated $\alpha$ during scaling of random allelic effect (Eq. 1). The simulation process was performed using R, PLINK[50] and MTG2[53]. The biasedness of estimates was assessed by Wald test.

### Determining scale factor ($\alpha$) across ancestries for LDAK-thin-$\alpha$ and GCTA-$\alpha$ model

We analysed six different anthropometric traits (BMI, standing height, waist circumference, hip circumference, waist hip ratio and weight) from the UK Biobank across different ancestries (White British, Other European, South Asian, African, and mixed ancestry cohorts). These traits were adjusted for demographic variables, UK biobank assessment centre, genotype measurement batch and population structure measured by the first 10 principal components (PCs)[34,54]. Demographic variable includes age, sex, year of birth, education, and Townsend deprivation index. Information of educational qualifications converted to education levels (years) for all the UK Biobank individuals[55].

GCTA-$\alpha$ and LDAK-thin-$\alpha$ models[23] were used to determine optimal $\alpha$ values for each of the ancestry groups. We considered various $\alpha$ values, e.g. $\alpha = -1, -0.875, -0.75, -0.675, -0.5, -0.375, -0.25, -0.125, 0$ and $0.125$, following the approach of Speed et al[21]. All GRMs with various $\alpha$ values were estimated using LDAK software[20], which set an equal weight to all SNPs in GCTA-$\alpha$ model and different weights to SNPs according to their LD scores in LDAK-thin-$\alpha$ model. Using GRMs,

SNP-based heritabilities were estimated for six anthropometric traits, using a multivariate linear mixed model[53] that fit six anthropometric traits simultaneously. Note that in the multivariate model, we treated the six traits independent without considering residual and genetic correlations between traits. This was because the optimal alpha value should be trait-specific as our main analysis was to estimate trait-specific cross-ancestry genetic correlation, i.e., equal weights from all six traits to obtain the optimal $\alpha$ value for each ancestry group. Finally, we identified optimal $\alpha$ that gives lowest AIC values, i.e., $AIC = 2k - 2\ln(L)$ where $\ln(L)$ is the logarithm of the maximum likelihood from the model and k is the number of parameters.

### Genetic correlation estimation using existing methods

Bivariate GREML[16,28,56,57] analyses were used to estimate heritability and cross-ancestry genetic correlation. In the analyses, we used four existing GRM methods (Table 1) to assess their performance, compared to the proposed method using simulated phenotypes, using PLINK[50] (-make-grm-gz for command GRM1 and GRM2) and GCTA[27] software (-sub-popu command for GRM3 and GRM4). The distribution of diagonal elements and off-diagonal elements across ancestries represented in Supplementary Figs. 6–10 (using $\alpha = -0.5$ and estimated in PLINK2). We also used Popcorn[2,16], which could be based on GWAS summary statistics. For these methods, we calculated empirical SE and its 95% confidence interval (CI) over 500 replicates to assess the unbiased estimation of the methods for each combination of ancestry pairs.

### Reporting summary

Further information on research design is available in the Nature Portfolio Reporting Summary linked to this article.

## Data availability

The genotype and phenotype data of the UK Biobank can be accessed through procedures described on its webpage (https://www.ukbiobank.ac.uk/). Source data are provided with this paper.

## Code availability

Source code of mtg2 can be accessed from https://github.com/mommy003/XA_GRM[58]. Simulated phenotypic data can be reproduced using R script available in https://github.com/mommy003/XA_GRM[58].

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

## Acknowledgements

This research is supported by the Australian Research Council (DP190100766) and the software development is supported by Cooperative Research Programme for Agriculture Science and Technology Development (PJ0160992021) from the Rural Development Administration, Republic of Korea. We are grateful to Professors Peter Visscher and Doug Speed for their constructive criticisms and comments on the manuscript. We thank the staff and participants of the UK Biobank for their important contributions. The analyses were performed using computational resources provided by the Australian Government through Gadi under the National Computational Merit Allocation Scheme (NCMAS), and HPCs (Tango and Statgen server) managed by UniSA IT.

## Author contributions

S.H.L. and M.M.M. conceived the idea. S.H.L. supervised the study. M.M.M. performed the analysis. S.H.L. and S.L. implemented computational functions in software. M.M.M. and S.H.L. verified the theory and analytical methods. M.M.M., J.S. and B.T. performed quality control of the data. B.B. provided critical feedback and key elements in interpreting the results. S.H.L. and M.M.M. wrote the first draft of the manuscript. All authors provided critical feedback and suggestions. All the authors contributed to editing and approval of the final manuscript.

## Competing interests

The authors declare no competing interests.

## Ethics approval

We used data from the UK Biobank (https://www.ukbiobank.ac.uk), the scientific protocol of which has been reviewed and approved by the Northwest Multi-centre Research Ethics Committee, National Information Governance Board for Health & Social Care, and Community Health Index Advisory Group. UK Biobank has obtained informed consent from all participants. Our access to the UK Biobank data was under the reference number 14575. The research ethics approval of this study has been obtained from the University of South Australia Human Research Ethics Committee.
