## [Peer Review File · Nature Communications]

A method for an unbiased estimate of cross-ancestry genetic correlation using individual-level dataReviewers' comments:

Reviewer #1 (Remarks to the Author):

In this manuscript, the authors proposed a new method to estimate cross-ancestry genetic correlation from individual genotype and phenotype data. Their new estimator of cross-ancestry genetic correlation can handle minor allele frequency dependent genetic architectures in different populations. Through simulations, they demonstrated that their method yielded unbiased estimates of cross-ancestry genetic correlation and was more robust to ancestry-specific genetic architectures than existing methods. They then applied their methods to estimate cross-ancestry genetic correlation in UK Biobank and found that cross-ancestry genetic correlation was significantly less than 1 for obesity, implicating heterogeneous genetic architectures in different populations. While I commend the authors for their effort in understanding genetic architectures in different ancestries, I believe the manuscript lacks important details regarding what is being estimated as well as additional insights into the genetic architectures of complex human traits in different ancestries.

Major comment:

The authors proposed a new method to estimate cross-ancestry genetic correlation. However, it's not crystal clear from the manuscript which version of genetic correlation is being estimated. Does the method estimate correlation of per-allele or standardized effect sizes in different ancestries? It seems the authors are interested in estimating cross-ancestry genetic correlation of standardized effect sizes, based on the equations provided in the manuscript. If that's the case, the authors should clearly define what they aim to estimate and provide justifications as to why they elected to estimate this particular version of cross-ancestry genetic correlation.

This manuscript also lacks some important details. For instance, which set of SNPs do the estimated genetic correlations pertain to? And how did the authors handle genetic variants present in one population but absent in the other population? Based on the Methods section, it seems the authors estimated cross-ancestry genetic correlation restricting to SNPs with MAF greater than 1% in both populations. If that's the case, the authors should clearly state this early in their manuscript.

It seems the authors used genetic variants present in both populations in their simulations. However, in reality, genetic variants private to a population may also impact diseases or complex traits. It would benefit the manuscript if the authors could demonstrate whether their method is also unbiased when ancestry-specific variants (variants present in one population but entirely absent in the other population) is included in the simulations.

It would benefit the manuscript if the authors could elaborate more on equation (2) on line 119. It seems this equation is crucial to the method. So, it would be critical that the authors discuss to a greater extent how their method differs from existing methods and why calculating relatedness using equation (2) yielded unbiased estimates of genetic correlation when there is MAF-dependent genetic architectures in different populations. The amount of text dedicated to equation (2) seems quite limited in the manuscript.

It's a bit unexpected and quite surprising that the estimated alphas were quite different across different ancestries, ranging from -0.125 for other Europeans to -0.825 for mixed ancestry cohorts. This result is a bit counterintuitive, as it would suggest that the impact of negative selection on effect size of genetic variants are highly ancestry-specific, which doesn't seem to comply with known biology – even though there's heterogeneity in the genetic architectures across ancestries, the underlying biology should be mostly consistent. The authors should elaborate more on why the estimated alphas were quite different across different ancestries, even within Europeans (i.e., white British vs. other Europeans).

The authors made some strong statement in Introduction without sufficient support. For instance, in the sentence spanning line 39 to 49, they authors made the claim that “effects...are dynamically distributed across populations”. This is likely an overstatement – even though genetic architectures are heterogeneous across populations, there’s probably a lot more homogeneity than heterogeneity. I would suggest that the authors consider revising this sentence and providing more references.

The authors also made an oversimplified regarding cross-population PRS in the sentence spanning line 55 to 57. The issues with PRS and cross-population PRS are more than just accuracy of estimated effect sizes. The authors should consider revising this sentence to include other issues such as MAF, LD, environmental effects, etc.

The authors compared their method with Popcorn, which is a method to estimate cross-ancestry genetic correlation from GWAS summary statistics data. While Popcorn might not be able to model MAF-dependent genetic architectures, a new method, called S-LDXR (Shi et al. 2021 Nat Commun), might be able to handle MAF-dependent genetic architectures. I would suggest the authors to briefly discuss S-LDXR in their manuscript. Additionally, it’s also important to clarify how the authors performed Popcorn analysis in their simulations.

The authors applied their method on several anthropometric traits. Although these results are interesting, most of them have already been reported in previous publications (e.g., Martin et al. 2019 Nat Genet).

Minor comments:

It would be great if the authors could clarify what GRM1, GRM2, GRM3, and GRM4 correspond to in the figure legend. This would make it much easier to understand figure 2.

Line 236: the p-value seems extremely low for such a large standard error.

Reviewer #3 (Remarks to the Author):

In this manuscript by Momin et al., the authors propose a method for the generation of less biased cross-ancestry genetic correlations. They assess the performance of their method with multiple scaling factor options, and iron out a more appropriate scaling factor for anthropometric traits for several sets of empirical populations. Finally they test their method on simulated phenotypes as well several anthropometric phenotypes from the UK Biobank. Their proposed method is interesting, however it is not yet clear to this reviewer that it outperforms other competing methods. There is a need for additional benchmarking against existing models, as well as a fuller discussion of what makes this method unique and improved as compared to other existing methods that also claim to produce unbiased cross-ancestry genetic correlations. I additionally have several concerns/suggested revisions regarding the analyses described here, which I detail below. In summary, I think that this manuscript would require substantial expansion to be suitable for publication in Nature Communications.

Major Comments:

1. As mentioned above, I feel a more complete description and assessment of the existing tools in this space and how they compare to the proposed model is needed. For example, the authors do include a test using POPCORN, but do not describe the actual model and how it differs. There is no discussion of any other existing methods, while several exist: the Multi-ancestry Meta Analysis or MAMA (Turley et al. BioRxiv) also purports to produce unbiased estimates of genetic correlations across ancestries by directly modeling LD and allele frequencies to account for heterogeneity in marginal effect sizes in

GWAS sumstats across populations. Cai et al. 2021 in AJHG, too describe a novel way to generate cross-population genetic correlations with the ultimate downstream goal of improving PRS. These and other methods' performance are not mentioned or benchmarked here and would be a valuable additional comparison point to this manuscript to justify this novel method's utility.

2. There is currently no benchmarking of computational efficiency across this and competing methods. This is an additional important consideration for downstream users, and some discussion of runtime/computational load required for various methods would be valuable, particularly as this method requires individual level data but some others can use sumstats.

3. During their tests with POPCORN, it is unclear if the parameters that would have been optimal to running that method were used. POPCORN directly takes in LD estimates from a reference panel, but here it appears that only blanket scaling metrics were used. Testing POPCORN's performance when running it in the best way for it would be the most fair comparison to running your new method with its optimal parameters.

4. The authors run their method on 6 empirical phenotypes from the UK Biobank, however these 6 phenotypes are overlapping such that they are effectively testing fewer than 6 phenotypes with similar genetic architectures. Specifically, waist and hip circumference are tested in addition to waist-hip ratio, height and weight are tested as well as BMI, and both sets of phenotypes are themselves addressing similar measures of general body size/weight. There are many more unrelated phenotypes in the UK biobank, and the paper would be strengthened by confirmation that their method performs well on additional phenotypes, particularly phenotypes with different genetic architectures from those already tested.

5. The authors' iron out a more appropriate scaling factor for variants across ancestries, which I see as a main strength of this manuscript. This is a useful parameter to know for future implementation of methods that take in a directly.

6. The population categorizations used do not seem ideal. For example, 'Asian' is an extremely broad definition and could do to be separated into East Asian and South Asian at least to result in more homogeneous groupings. Similarly, the authors examine 'white British' as compared to all other Europeans; this appears less problematic based on their results but some justification that this differentiation makes sense would be helpful; e.g. Finns are routinely analyzed separately from other Europeans due to their notable divergence (gnomAD has reports for Finns vs Non-Finnish Europeans from this reason). And most notably, a 'mixed ancestry' cohort is included as a large catch-all set including Asian, African, Caribbean, and other admixed individuals. These individuals vary dramatically from one another, and will have effects of admixture LD that will impact results. As it is, there is no visualization of the population composition of each group nor is there discussion of how the stratification that is present in many of these groups will affect the results. Supp Fig 1 provides a PC visualization, but this appears to be the midpoints for each group, and itself highlights the large variation in ancestry within some of their groupings as compared to others. There is also one note in the Discussion regarding the model choice should depend on homogeneity of the cohort, but this seems like an important point that is currently underdeveloped.

7. The authors should clarify their definition and use of the term 'causal effects' throughout. At several points in the manuscript it appears the authors are actually referring to the marginal effect at tagging SNPs, but these effect estimates are referred to as causal. It should be made clear when the authors mean a true effect at a causal locus versus scaling the estimated effect at a tagging locus.

8. The authors argue that their method does better than competing methods in cases where there is a different underlying genetic architecture across populations, which in their definition primarily means a different underlying set of causal SNPs. It also appears that this is their assumed model for traits; e.g. second sentence of their manuscript "In humans, causal loci and their effects on complex traits are dynamically distributed across populations..." How often is this actually the case? Prior work has generally shown that causal effects tend to be shared across populations (Peterson 2019, Lam 2019, Marigorta 2013).

Minor comments:

a. The language could be tightened up, I noticed some typos throughout.

b. Please add page and line numbers.

Reviewers' comments:

Reviewer #1 (Remarks to the Author):

In this manuscript, the authors proposed a new method to estimate cross-ancestry genetic correlation from individual genotype and phenotype data. Their new estimator of cross-ancestry genetic correlation can handle minor allele frequency dependent genetic architectures in different populations. Through simulations, they demonstrated that their method yielded unbiased estimates of cross-ancestry genetic correlation and was more robust to ancestry-specific genetic architectures than existing methods. They then applied their methods to estimate cross-ancestry genetic correlation in UK Biobank and found that cross-ancestry genetic correlation was significantly less than 1 for obesity, implicating heterogenous genetic architectures in different populations. While I commend the authors for their effort in understanding genetic architectures in different ancestries, I believe the manuscript lacks important details regarding what is being estimated as well as additional insights into the genetic architectures of complex human traits in different ancestries.

R: We appreciate the reviewer for their praise, “I commend the authors for ...”. For the concern of lack of details, we have added the details in the overview section (in an earlier part of the text) so that it is now clearer and more visible (please see point-by-point response below).

Major comment:

The authors proposed a new method to estimate cross-ancestry genetic correlation. However, it's not crystal clear from the manuscript which version of genetic correlation is being estimated. Does the method estimate correlation of per-allele or standardized effect sizes in different ancestries? It seems the authors are interested in estimating cross-ancestry genetic correlation of standardized effect sizes, based on the equations provided in the manuscript. If that's the case, the authors should clearly define what they aim to estimate and provide justifications as to why they elected to estimate this particular version of cross-ancestry genetic correlation.

R: We thank the reviewer for this comment. The proposed method estimates the correlation of per-allele effect size (original scale) while most existing methods estimate the correlation of per-allele effect size modified by a function of allele frequency and alpha that is the scale factor determining the genetic architecture, which is briefly explained in the following (more details in Methods).

The heritability can be written as

$$h^2 = \sigma_g^2 = \sum_{i=1}^M \text{var}(v_i)$$

where $\text{var}(v_i)$ is the genetic variance of the i^{th} causal variant ($i=1-M$). Following Falconer and Mackay (1996), the genetic variance of the i^{th} causal variant is $\text{var}(v_i) = 2p_i(1 - p_i)\gamma_i^2$

where $\gamma_i = \beta_i \times [2p_i(1 - p_i)]^\alpha$ as suggested by previous studies (reference # 21, 22, 24) and β_i is per-allele effect size at the i^{th} variant, noting that γ_i is the allele effect size scaled by a function of its frequency and alpha (probably due to selection, interaction, linkage disequilibrium, population stratification, genetic drift etc.).

What we are concern is β_i , i.e. the correlation of per-allele effect size in the original scale. Therefore, we propose to correctly account for alpha in the estimation of GRM (eq. 2), which can disentangle β_i from γ_i . (This detail is now added in line 98-99, line 106-107, line 134 - 138 and line 402-406, 418-419 in Methods).

This manuscript also lacks some important details. For instance, which set of SNPs do the estimated genetic correlations pertain to? And how did the authors handle genetic variants present in one population but absent in the other population? Based on the Methods section, it seems the authors estimated cross-ancestry genetic correlation restricting to SNPs with MAF greater than 1% in both populations. If that's the case, the authors should clearly state this early in their manuscript.

R: We clearly stated as, "Our main aim is to unbiasedly estimate cross-ancestry genetic correlation for a complex trait, using common SNPs (e.g., those with MAF > 0.01) presented for both populations" which is now in the first section in Results (line 90). We made it clear that MAF is greater than 1% (line 91 and 148). In this study, we used HapMap3 SNP to estimate cross-ancestry genetic correlation, as they are robust and reliable for genetic analysis, which is explicitly mentioned in the text (page 22, line number 534-535).

It seems the authors used genetic variants present in both populations in their simulations. However, in reality, genetic variants private to a population may also impact diseases or complex traits. It would benefit the manuscript if the authors could demonstrate whether their method is also unbiased when ancestry-specific variants (variants present in one population but entirely absent in the other population) is included in the simulations.

R: We thank the reviewer for this insightful comment. We agree that the source of the genetic variance is heterogenous (e.g. different mutations and causal effects) across ancestries. However, our main interest is to estimate the correlation between common genetic variants between ancestries as clearly shown in the text (line 90). For the common variants, there can also be a substantial heterogeneity in their causal effects, i.e. SNPs with positive allele substitution effects in one ancestry group can have negative allele substitution effects in another ancestry group. We tested various simulation scenarios including such a case (e.g. $r_G=0$ in Figure 3) where a substantial proportion of SNPs of which per allele effect size is negative ($|z\text{-score}| < -0.5$) in one population but positive ($|z\text{-score}| > 0.5$ in another population (see Table R1 below). In any case, the proposed method could also provide unbiased estimates (Figure 3).

Table R1. Mean of percentages of SNP that has negative (z -score < -0.5) size in one population but positive (z -score > 0.5) in another population

Ancestries	$r_g = 0$	$r_g = 0.5$	$r_g = 1$
	Mean%±SE	Mean%±SE	Mean%±SE
British vs African	30.81±0.003	11.82±0.002	0
British vs South Asian	31.01±0.003	11.73±0.002	0
South Asian vs African	30.72±0.003	11.84±0.002	0

In this simulation, we used 1000 SNP to simulate causal effect size for two ancestries for each pair. Mean of percentages based on 100 replicates (SE is empirical SE for 100 replicates)

It would benefit the manuscript if the authors could elaborate more on equation (2) on line 119. It seems this equation is crucial to the method. So, it would be critical that the authors discuss to a greater extent how their method differs from existing methods and why calculating relatedness using equation (2) yielded unbiased estimates of genetic correlation when there is MAF-dependent genetic architectures in different populations. The amount of text dedicated to equation (2) seems quite limited in the manuscript.

R: We appreciate for the comment and added an elaboration in line 134 as “Eq. 2 can account for heterogenous α across ancestries so that per allele effect size can be estimated accurately in the original scale in each ancestry, therefore, the correlation of per allele effect size is unbiasedly estimated, i.e. disentangling β_i from γ_i . However, existing methods, not accounting for alpha, cannot disentangle β_i from γ_i . Please also see pages 17 - 21. We also elaborated this in line 402-406, 417-419.

It’s a bit unexpected and quite surprising that the estimated alphas were quite different across different ancestries, ranging from -0.125 for other Europeans to -0.825 for mixed ancestry cohorts. This result is a bit counterintuitive, as it would suggest that the impact of negative selection on effect size of genetic variants are highly ancestry-specific, which doesn’t seem to comply with known biology – even though there’s heterogeneity in the genetic architectures across ancestries, the underlying biology should be mostly consistent. The authors should elaborate more on why the estimated alphas were quite different across different ancestries, even within Europeans (i.e., white British vs. other Europeans).

R: We agree that the underlying biology is mostly consistent across ancestries at the individual gene level. However, the scale factor (alpha) approximates the genetic architecture involving many genes (tagged by genome-wide SNPs), their allele substitution effects and allele frequencies across the genome, possibly including gene x gene and gene x ancestry effects. There are several studies that support such heterogeneity, i.e.

Sirugo et al.[1] reported that the most common causative allele ($\Delta F508$) in the *CFTR* gene accounts for more than 70% of cystic fibrosis cases in European, whereas it accounts for only 29% of cases in people of the African people.

Studies have shown that in Europeans, the proportion of variance in drug metabolism explained by SNPs in the *CYP2C9*, *VKORC1*, and *CYP4F2* genes is 18%, 30%, and 11%, respectively [2]; however, in patients of African descent these variants explain much less of the differences in drug metabolism.

PRS may not be transferable across diverse populations. Indeed, inconsistencies in the directions of effect of risk variants have been observed across ethnic groups [3]. This was confirmed by the follow-up study by Martin et al. [4].

We incorporated this explanation with these references in the main text (line 39-40, 53-54)

For white British and other Europeans, the estimated alpha values are not significantly different to each other (-0.25 Vs. -0.125, Delta AIC is <2) (Figure 1 and Supplementary Figure 2).

The authors made some strong statement in Introduction without sufficient support. For instance, in the sentence spanning line 39 to 49, they authors made the claim that “effects...are dynamically distributed across populations”. This is likely an overstatement – even though genetic architectures are heterogeneous across populations, there’s probably a lot more homogeneity than heterogeneity. I would suggest that the authors consider revising this sentence and providing more references.

R: We revised the sentence as “effects on complex traits are not uniformly distributed across populations” (line 39, page 2). We added three more references in lines between 39 to 49. We also changed the word “dynamically” throughout the text (line 285, 288 and 419)

The authors also made an oversimplified regarding cross-population PRS in the sentence spanning line 55 to 57. The issues with PRS and cross-population PRS are more than just accuracy of estimated effect sizes. The authors should consider revising this sentence to include other issues such as MAF, LD, environmental effects, etc.

R: We thank the reviewer for this comment and revised as “... polygenic risk prediction depends on the accuracy of estimated SNP associations, minor allele frequencies (MAF), linkage disequilibrium (LD) and environmental heterogeneity, causing a disparity in genetic prediction across populations” (page2, line 57-58)

The authors compared their method with Popcorn, which is a method to estimate cross-ancestry genetic correlation from GWAS summary statistics data. While Popcorn might not be able to model MAF-dependent genetic architectures, a new method, called S-LDXR (Shi et al. 2021 Nat Commun), might be able to handle MAF-dependent genetic architectures. I would suggest the authors to briefly discuss S-LDXR in their manuscript. Additionally, it’s also important to clarify how the authors performed Popcorn analysis in their simulations.

R: We appreciate the reviewer for this comment, and we added a discussion about S-LDXR in the main text (line 333-337).

Re. Popcorn. We have estimated LD score for each pair of ancestries, following the Popcorn manual (<https://github.com/brielin/popcorn>). Note that we estimated linkage disequilibrium (LD) scores for each pair of ancestries using in-sample (i.e. UK Biobank), which would give the best performance of summary statistics-based methods [5, 6]. This is an ideal situation for Popcorn, i.e. individual level genotypes are available for popcorn to estimate LD scores. The popcorn command we used to compute LD scores for two populations was

```
popcorn compute -v 1 --bfile1 ancestry1 --bfile2 ancestry2 scores.txt.
```

GWAS summary statistics for each ancestry group was separately estimated by PLINK2, according to Popcorn manual. Using the summary statistics and LD scores, we used the following Popcorn command to estimate cross ancestry genetic correlations.

```
popcorn fit -v 1 --cfile scores.txt --gen_effect --sfile1  
ancestry1_sumstats.txt --sfile2 ancestry2_sumstats.txt  
correlation_output.txt
```

It is noted that Popcorn's performance became poorer when using LD scores estimated from 1KG reference panel (result not shown).

We added this detail in the main text (line 260-273 in Supplementary Notes).

The authors applied their method on several anthropometric traits. Although these results are interesting, most of them have already been reported in previous publications (e.g., Martin et al. 2019 Nat Genet).

R: Martin et al.[4] reported the difference between R^2 values based on ancestry specific PRS. This predictive ability of PRS is a function of heritability within each ancestry, effective number of chromosome segments, genetic correlation between ancestries, and sample size. Martin et al. is different from our study that has reported estimated genetic correlations between ancestries (White British, Other Europeans, South Asian and African ancestries), using the proposed method.

We are now preparing for an application paper in which we apply our proposed method to cardiovascular and metabolic traits. The results for pulse rate, cholesterol, LDL cholesterol and education are shown in Figure R1. There are some interesting observations, e.g. for cholesterol and education, there is significant genetic heterogeneity between ancestries. Although we do not add this result in the current manuscript, this shows that the proposed method performs well on additional traits that will be published in a subsequent application paper (the subsequent application paper will be submitted soon).

Figure R1. Estimated cross-ancestry genetic correlations for pulse rate, cholesterol, LDL-cholesterol, and education.

Minor comments:

It would be great if the authors could clarify what GRM1, GRM2, GRM3, and GRM4 correspond to in the figure legend. This would make it much easier to understand figure 2.

R: Clarified in Figure 2 and Figure 3 legend (for more details please see in Table 1)

Line 236: the p-value seems extremely low for such a large standard error.

R: Typo and corrected in the main text (line 243)

Reviewer #3 (Remarks to the Author):

In this manuscript by Momin et al., the authors propose a method for the generation of less biased cross-ancestry genetic correlations. They assess the performance of their method with multiple scaling factor options, and iron out a more appropriate scaling factor for anthropometric traits for several sets of empirical populations. Finally, they test their method on simulated phenotypes as well several anthropometric phenotypes from the UK Biobank. Their proposed method is interesting; however it is not yet clear to this reviewer that it outperforms other competing methods. There is a need for additional benchmarking against existing models, as well as a fuller discussion of what makes this method unique and improved as compared to other existing methods that also claim to produce unbiased cross-ancestry genetic correlations. I additionally have several concerns/suggested revisions regarding the analyses described here, which I detail below. In summary, I think that this manuscript would require substantial expansion to be suitable for publication in Nature Communications.

R: We thank the reviewer for this comment. We additionally compare some more existing methods including their unbiasedness and computational efficiency (see point-by-point response below). We also have made it clear what is being estimated by the proposed method, compared to existing methods, which has been addressed in the response to the first reviewer's question, i.e.

The proposed method estimates the correlation of per-allele effect size (original scale) while most existing methods estimate the correlation of per-allele effect size modified by a function of allele frequency and alpha that is the scale factor determining the genetic architecture, which is briefly explained in the following (more details in Methods). The heritability can be written as

$$h^2 = \sigma_g^2 = \sum_{i=1}^M \text{var}(v_i)$$

where $\text{var}(v_i)$ is the genetic variance of the i^{th} causal variant ($i=1-M$). Following Falconer and Mackay (1996), the genetic variance of the i^{th} causal variant is

$$\text{var}(v_i) = 2p_i(1 - p_i)\gamma_i^2$$

where $\gamma_i = \beta_i \times [2p_i(1 - p_i)]^\alpha$ as suggested by previous studies (reference # 21, 22, 24) and β_i is per-allele effect size at the i^{th} variant, noting that γ_i is the allele effect size scaled by a function of its frequency and alpha (probably due to selection, interaction, linkage disequilibrium, population stratification, genetic drift etc.).

What we are concern is β_i , i.e. the correlation of per-allele effect size in the original scale. Therefore, we propose to correctly account for alpha in the estimation of GRM (eq. 2), which can disentangle β_i from γ_i . (This detail is now added in line 98-99, line 106-107, line 134 - 138 and line 402-406, 418-419 in Methods).

Major Comments:

1. As mentioned above, I feel a more complete description and assessment of the existing tools in this space and how they compare to the proposed model is needed. For example, the authors do include a test using POPCORN, but do not describe the actual model and how it differs. There is no discussion of any other existing methods, while several exist: the Multi-

ancestry Meta Analysis or MAMA (Turley et al. BioRxiv) also purports to produce unbiased estimates of genetic correlations across ancestries by directly modeling LD and allele frequencies to account for heterogeneity in marginal effect sizes in GWAS sumstats across populations. Cai et al. 2021 in AJHG, too describe a novel way to generate cross-population genetic correlations with the ultimate downstream goal of improving PRS. These and other methods' performance are not mentioned or benchmarked here and would be a valuable additional comparison point to this manuscript to justify this novel method's utility.

R: We thank the reviewer for this comment. Our main interest is to provide unbiased estimates of genetic correlation between ancestry groups using individual level data, which is likely to provide more accurate estimates although it is computationally demanding, compared to GWAS summary statistics-based methods [5, 7]. Indeed, the Popcorn software paper [8] compared the performance of Popcorn with an individual level data-based method (i.e. GCTA) as the gold standard. It stated, “We found that GCTA and Popcorn agreed on the global distribution of heritability and that GCTA’s estimates of genetic correlation had a similar distribution to Popcorn’s estimates of genetic-effect and genetic-impact correlation” (their page 82 in Brown et al.[8]).

The gold standard method was explicitly evaluated in our study (named as GRM1 in Figure 2-3). We compare the performance of the proposed method with the gold standard method with various settings (i.e. GRM1 – 4 in Figure 3), which clearly demonstrates that the proposed method outperforms existing methods. Additionally, we also show here that the proposed method outperforms GWAS summary statistics-based methods (Popcorn and XPASS) according to the reviewers’ comments (see Table R2). Please note that MAMA is equivalent to popcorn in terms of estimated genetic correlations, and it did not provide SE of estimate, so we didn’t include MAMA in Table R2.

For Popcorn, we estimated LD scores for each pair of ancestries, following the Popcorn manual (<https://github.com/brielin/popcorn>). Note that we estimated LD scores for each pair of ancestries using in-sample (i.e. UK Biobank), which would give the best performance of summary statistics-based methods [5, 6]. This is an ideal situation for Popcorn, i.e. individual level genotypes are available for popcorn to estimate LD scores.

The popcorn command we used to compute LD scores for two populations was

```
popcorn compute -v 1 --bfile1 ancestry1 --bfile2 ancestry2 scores.txt.
```

GWAS summary statistics for each ancestry group was separately estimated by PLINK2, according to Popcorn manual. Using the summary statistics and LD scores, we used the following Popcorn command to estimate cross ancestry genetic correlations.

```
popcorn fit -v 1 --cfile scores.txt --gen_effect --sfile1  
ancestry1_sumstats.txt --sfile2 ancestry2_sumstats.txt  
correlation_output.txt
```

We added this detail as well as that for XPASS (Cai et al. 2021) (in line 260-304 in supplementary notes).

We explicitly checked and compared the estimation among Popcorn [8], XPASS [9] and proposed method with a larger sample size (10,000 for a pair of ancestries). It is shown that estimates of Popcorn and XPASS were biased whereas the proposed method provide unbiased estimates (added in line 208 – 214 in the main text and Supplementary Table 13).

Table R2. Comparing heritability and cross ancestry genetic correlation Popcorn, XPASS and Proposed method using estimated scale factor

True values	Combinations of ethnicities	Model using estimated scaling factor ($\alpha = -0.25, -0.625$ and -0.75 respectively, for white British, Asian and African)								
		Popcorn			XPASS			Proposed method		
		Estimated h_{eth1}^2	Estimated h_{eth2}^2	Estimated r_g	Estimated h_{eth1}^2	Estimated h_{eth2}^2	Estimated r_g	Estimated h_{eth1}^2	Estimated h_{eth2}^2	Estimated r_g
$h_{eth1}^2 = 0.5$	White British and African	0.46±0.016	0.13±0.019	0.36±0.051	0.53±0.011	0.65±0.069	0.55±0.028	0.497±0.008	0.508±0.008	0.505±0.018
$h_{eth2}^2 = 0.5$	White British and Asian	0.47±0.016	0.37±0.022	0.41±0.044	0.55±0.010	0.46±0.023	0.54±0.017	0.496±0.008	0.501±0.009	0.510±0.013
$r_g = 0.5$	Asian and African	0.41±0.014	0.31±0.025	0.40±0.049	0.50±0.043	0.53±0.067	0.58±0.034	0.495±0.008	0.494±0.009	0.514±0.019

Simulation was based on 1000 random common SNPs as causal 10,000 individuals (5000 from each ancestry). All the estimated value (heritability and trans-ethnic genetic correlation) in the table based 50 replications. h_{eth1}^2 and h_{eth2}^2 indicates heritability of first and second trait in combined population. For simulation true heritability was 0.5 for both ethnic group and true genetic correlation was also 0.50 in each pair of bivariate simulation. Red indicates biased estimation.

It is noted that Popcorn's performance became poorer when using LD scores estimated from 1KG reference panel (result not shown).

2. There is currently no benchmarking of computational efficiency across this and competing methods. This is an additional important consideration for downstream users, and some discussion of runtime/computational load required for various methods would be valuable, particularly as this method requires individual level data but some others can use sumstats.

R: Thanks for this comment. We also benchmark the computational efficiency of the methods (Table R3). The proposed method requires individual level data, which is expected to be computationally more demanding, compared to those based on summary stats. As expected, the computational efficiency of Popcorn is much higher than the proposed method. However, we believe that the accuracy of estimates (unbiasedness) is equally or more important as shown in Table R2. Nonetheless, Table R3 shows that the proposed method can be compatible with XPASS. These results were added (line 211 in the main text and Supplementary Table 14).

Table R3. We compare the speed among Popcorn, XPASS and our proposed method (see following table).

Conditions	Popcorn	XPASS	Proposed method
(1000+1000) individual and ~200k common SNP	For score: 1 min 46 sec For GWAS: 30sec For estimation: 39 sec RAM: 0.34GB	For GWAS: 30 sec For estimation: 4 min 19 sec RAM: 16G	For GRM: 1 min 37 sec For GREML: 15 sec RAM: 926 MB
(5000+5000) individual and ~200k common SNP	For score: 4min 31 sec For GWAS: 2 min For estimation: 42 sec RAM: 0.36 GB	For GWAS: 2 min For estimation: 26 min RAM: 54G	For GRM: 17 min 59 sec For GREML: 4 min 21 sec RAM: 22G
(10000+10000) individual and ~200k common SNP	For score: 7 min 39 sec For GWAS: 2 min 12 sec For estimation: 45 sec RAM: 0.37GB	For GWAS: 2 min 12 sec For estimation: 1 hr 22 min RAM: 76G	For GRM: 58 min 05 sec For GREML: 27 min 9 sec RAM: 52G

Using a single CPU (2.0 GHz).

Proposed method and XPASS can use a parallel computing and their computational efficiency can be further increased. Parallel computing is not available for Popcorn.

3. During their tests with POPCORN, it is unclear if the parameters that would have been optimal to running that method were used. POPCORN directly takes in LD estimates from a reference panel, but here it appears that only blanket scaling metrics were used. Testing POPCORN's performance when running it in the best way for it would be the most fair comparison to running your new method with its optimal parameters.

R: Please see the response to the first comment above. According to the Popcorn manual (<https://github.com/brielin/popcorn>), we estimated LD scores for each pair of ancestries using in-sample (i.e. UK Biobank), which would give the best performance of summary statistics-based methods [5, 6]. This is an ideal situation for Popcorn, i.e. individual level genotypes are available for popcorn to estimate LD scores (Supplementary Table 13 and Supplementary Notes). We also used LD scores from 1KG reference samples (not in-sample), which resulted in more biased estimates (result not shown).

4. The authors run their method on 6 empirical phenotypes from the UK Biobank, however these 6 phenotypes are overlapping such that they are effectively testing fewer than 6 phenotypes with similar genetic architectures. Specifically, waist and hip circumference are tested in addition to waist-hip ratio, height and weight are tested as well as BMI, and both sets of phenotypes are themselves addressing similar measures of general body size/weight. There are many more unrelated phenotypes in the UK biobank, and the paper would be strengthened by confirmation that their method performs well on additional phenotypes, particularly phenotypes with different genetic architectures from those already tested.

R: We think our simulations and real data analysis for the 6 anthropometric traits can have demonstrated the performance of proposed method, comparing with existing methods (both individual level- and summary statistics-based methods).

We are now preparing for an application paper in which we have applied our proposed method to cardiovascular and metabolic traits. The results for pulse rate, cholesterol, LDL cholesterol and education are shown in Figure R1. There are some interesting observations, e.g. for cholesterol and education, there is significant genetic heterogeneity between ancestries. Although we do not add this result in the current manuscript, this shows that the proposed method performs well on additional traits that will be published in a subsequent application paper (the subsequent application paper will be submitted soon).

Figure R1. Estimated cross-ancestry genetic correlations for pulse rate, cholesterol, LDL-cholesterol, and education.

5. The authors' iron out a more appropriate scaling factor for variants across ancestries, which I see as a main strength of this manuscript. This is a useful parameter to know for future implementation of methods that take in α directly.

R: We thank the reviewer for this positive comment.

6. The population categorizations used do not seem ideal. For example, 'Asian' is an extremely broad definition and could do to be separated into East Asian and South Asian at least to result in more homogeneous groupings. Similarly, the authors examine 'white British' as compared to all other Europeans; this appears less problematic based on their results but some justification that this differentiation makes sense would be helpful; e.g. Finns are routinely analyzed separately from other Europeans due to their notable divergence (gnomAD has reports for Finns vs Non-Finnish Europeans from this reason). And most notably, a 'mixed ancestry' cohort is included as a large catch-all set including Asian, African, Caribbean, and other admixed individuals. These individuals vary dramatically from one another, and will have effects of admixture LD that will impact results. As it is, there is no visualization of the population composition of each group nor is there discussion of how the stratification that is present in many of these groups will affect the results. Supp Fig 1 provides a PC visualization, but this appears to be the midpoints for each group, and itself highlights the large variation in ancestry within some of their groupings as compared to others. There is also one note in the Discussion regarding the model choice should depend on homogeneity of the cohort, but this seems like an important point that is currently underdeveloped.

R: The 'Asian' is actually South Asian (now revised). We did not consider East Asian because of a small sample size in UK Biobank ($n \approx 1400$), which lacks power. We also added an explanation why White British and Other Europeans were grouped separately (line 524-525). The mixed ancestry cohort is an admixed population including multiple

ancestry groups, and there is no intention to group it as an ancestry. We made this clearer in the main text (line 86, 518-520, 523-524). We also added a box including ~95% of the population samples (mean \pm 2SD for PC1 and PC2) into the Supplementary Figure 1.

7. The authors should clarify their definition and use of the term ‘causal effects’ throughout. At several points in the manuscript, it appears the authors are actually referring to the marginal effect at tagging SNPs, but these effect estimates are referred to as causal. It should be made clear when the authors mean a true effect at a causal locus versus scaling the estimated effect at a tagging locus.

R: We thank the reviewer for this comment. We have revised from “causal effects” to “allelic effects” throughout the text (Abstract, Line 114, Line 177, Line 286, Line 399, Line 549, Line 550, Line 553, Line 556, Line 564, Conclusion). We have also clarified that the proposed method estimates the correlation of per-allele effect size (original scale) while most existing methods estimate the correlation of per-allele effect size modified by a function of allele frequency and alpha that is the scale factor determining the genetic architecture (This detail is now added in line 98-99, line 106-107, line 134 - 137 and line 402-405, 418-419 in Methods). Please also see our response to the first question from the reviewer #1.

8. The authors argue that their method does better than competing methods in cases where there is a different underlying genetic architecture across populations, which in their definition primarily means a different underlying set of causal SNPs. It also appears that this is their assumed model for traits; e.g. second sentence of their manuscript “In humans, causal loci and their effects on complex traits are dynamically distributed across populations...” How often is this actually the case? Prior work has generally shown that causal effects tend to be shared across populations (Peterson 2019, Lam 2019, Marigorta 2013).

R: We revised the second sentence “causal loci and their effects on complex traits are not uniformly distributed across populations such that the same trait can be genetically heterogenous between two ancestry groups” (line 38-39).

We agree that the underlying biology is mostly consistent across ancestries at the individual gene level. However, the scale factor (alpha) approximates the genetic architecture involving many genes (tagged by genome-wide SNPs), their allele substitution effects and allele frequencies across the genome, possibly including gene x gene and gene x ancestry effects. There are several studies that support such heterogeneity, i.e.

Sirugo et al.[1] reported that the most common causative allele (Δ F508) in the *CFTR* gene accounts for more than 70% of cystic fibrosis cases in European, whereas it accounts for only 29% of cases in people of the African people.

Studies have shown that in Europeans, the proportion of variance in drug metabolism explained by SNPs in the *CYP2C9*, *VKORC1*, and *CYP4F2* genes is 18%, 30%, and 11%, respectively [2]; however, in patients of African descent these variants explain much less of the differences in drug metabolism.

PRS may not be transferable across diverse populations. Indeed, inconsistencies in the directions of effect of risk variants have been observed across ethnic groups [3]. This was confirmed by the follow-up study (Martin et al. [4])

We incorporated this explanation with these references in the main text (line 39-40, 53-54)

Minor comments:

a. The language could be tightened up, I noticed some typos throughout.

R: Typos were checked and corrected.

b. Please add page and line numbers.

R: Thanks. Page and line number has been added.

References

1. Sirugo, G., S.M. Williams, and S.A. Tishkoff, *The missing diversity in human genetic studies*. Cell, 2019. **177**(1): p. 26-31.
2. Johnson, J.A., et al., *Clinical Pharmacogenetics Implementation Consortium (CPIC) guideline for pharmacogenetics-guided warfarin dosing: 2017 update*. Clinical Pharmacology & Therapeutics, 2017. **102**(3): p. 397-404.
3. Martin, A.R., et al., *Human demographic history impacts genetic risk prediction across diverse populations*. The American Journal of Human Genetics, 2017. **100**(4): p. 635-649.
4. Martin, A.R., et al., *Clinical use of current polygenic risk scores may exacerbate health disparities*. Nature Genetics, 2019. **51**(4): p. 584-591.
5. Bulik-Sullivan, B.K., et al., *LD Score regression distinguishes confounding from polygenicity in genome-wide association studies*. Nature Genetics, 2015. **47**(3): p. 291-295.
6. Bulik-Sullivan, B., et al., *An atlas of genetic correlations across human diseases and traits*. Nature Genetics, 2015. **47**(11): p. 1236-1241.
7. Ni, G., et al., *Estimation of genetic correlation via linkage disequilibrium score regression and genomic restricted maximum likelihood*. The American Journal of Human Genetics, 2018. **102**(6): p. 1185-1194.
8. Brown, B.C., et al., *Transethnic genetic-correlation estimates from summary statistics*. The American Journal of Human Genetics, 2016. **99**(1): p. 76-88.
9. Cai, M., et al., *A unified framework for cross-population trait prediction by leveraging the genetic correlation of polygenic traits*. The American Journal of Human Genetics, 2021. **108**(4): p. 632-655.

REVIEWER COMMENTS

Reviewer #3 (Remarks to the Author):

Original Summary:

In this manuscript by Momin et al., the authors propose a method for the generation of less biased cross-ancestry genetic correlations. They assess the performance of their method with multiple scaling factor options, and iron out a more appropriate scaling factor for anthropometric traits for several sets of empirical populations. Finally they test their method on simulated phenotypes as well several anthropometric phenotypes from the UK Biobank.

Comments on revisions:

In response to my suggestions, the authors have now benchmarked computational efficiency, presented in a new Supplementary Table. While the results highlight that the high computational load required by their software may be a substantial burden for some analyses, they argue that their cost to speed from using individual level data is made up for by improvements to accuracy.

More description of the models is now provided in the text, as well as a fuller description of the comparison benchmarking done.

In response to my recommendation to benchmark additional unrelated phenotypes with differing genetic architectures, the authors provide some compelling figures in their Response to Reviewers. While not much detail is given about them, these results do appear to support the performance of their proposed model, however these additional phenotypes are not added to this manuscript. I am not sure I agree that the desire to save all other phenotypes to have a second applied publication can justify excluding any more here, as this article should serve as a proof of principle that their proposed method indeed can perform across the spectrum of polygenicity.

The authors have clarified their population label for the South Asian (formerly 'Asian') grouping. However, referring to the catch-all collection of all other ancestry individuals outside the tested primary labels as 'admixed' is not an appropriate terminology. Based on the description in the text, this does not seem to be a cohort comprising one population of admixed individuals but rather a collection of individuals from many different ancestry/population groups, some of whom are homogeneous. This grouping should be changed to 'mixed' throughout, as it is presented in their figures.

Reviewer #4 (Remarks to the Author):

Most of the comments of reviewer 1 were well addressed by the authors. I have some follow-up questions:

1. Following up the third question of review 1, Is the proposed method robust to model misspecification? For example, the proposed method assume an infinitesimal model where all the SNPs are causal SNPs. What if the true model is that only a small proportion of SNPs are causal? What if the causal SNPs are not 100% common for the populations.
2. "It is noted that the computational efficiency of the proposed method can be increased further with parallel computing." Would the authors please elaborate more on how does the parallel computing work for this method?
3. I was a reader of the authors' previous work,

<https://academic.oup.com/bioinformatics/article/28/19/2540/289604?login=true>. Would the authors please comment on the most improvements compared with other old method in the angle of modeling? (I know this method is used to estimate croon-population genetic correlation)

Reviewer #5 (Remarks to the Author):

The authors have addressed most comments, but there are still concerns regarding the proposed method and its use in realistic data settings. In general, the authors have not sufficiently shown that their method performs better than existing methods (via simulations).

1. It would be helpful to the reader for the author to concisely state how their proposed method differs from the existing methods, without the need for the reader to read 2 pages in the Methods. It appears that one reason the proposed method has gains over other methods is because of its use of individual-level data estimates (e.g. $\text{var}(x_i)$ instead of its expectation), which would be expected. If there is no varying alpha and summary-level data estimates are used is the proposed method the same as GRM4? More is needed in the main text to explain the difference between previous methods and the proposed method. This question was touched on by both Reviewers 1 and 3, but not satisfactorily addressed.

2. The following comment of Reviewer 1 (bottom page 2) lists an important issue that the authors have not sufficiently addressed in their response

“It seems the authors used genetic variants present in both populations in their simulations. However, in reality, genetic variants private to a population may also impact diseases or complex traits. It would benefit the manuscript if the authors could demonstrate whether their method is also unbiased when ancestry-specific variants (variants present in one population but entirely absent in the other population) is included in the simulations.”

Although the focus is on variants that have $\text{MAF} > .01$ in each population, the suggested realistic scenario should be considered in simulations. i.e. in simulations, allow for some causal variants that have $\text{MAF} > .01$ in one population and $\text{MAF} < .01$ in the other population. Then, apply methods as usual, using only variants with $\text{MAF} > .01$ in both populations.

3. The current simulation setting is tailored to the proposed method and robustness needs to be demonstrated. When the competing methods are used in simulation settings tailored to themselves, GRM3 and GRM4 perform well in each ancestry combination (e.g. $\alpha = -0.5$ in both simulation and estimation, Fig 2). However, in the simulation setting of Fig 3, ancestry-specific alpha values (more realistic) are used with these methods having $\alpha = -0.5$ and the proposed method having alpha matching the simulation settings, which is an unfair comparison.

a. For the Fig 3 setting, it would be informative to run simulations with varying alpha and to have different (from the simulation settings) varying alpha in the proposed method.

b. For the Fig 2 setting, the proposed method should also be included (i) using $\alpha = -0.5$ in the new method instead of varying alpha, to match the sim settings as for the other methods; (ii) using varying alpha in the method, as proposed for the new method.

These are the minimum simulation comparisons and as the authors are introducing a new method, a more extensive simulation study, with realistic settings, is needed.

Response Letter

Reviewer #3:

1. In this manuscript by Momin et al., the authors propose a method for the generation of less biased cross-ancestry genetic correlations. They assess the performance of their method with multiple scaling factor options, and iron out a more appropriate scaling factor for anthropometric traits for several sets of empirical populations. Finally, they test their method on simulated phenotypes as well several anthropometric phenotypes from the UK Biobank.

Author Response: We thank the reviewer for this nice summary of the main findings in this study.

2. In response to my suggestions, the authors have now benchmarked computational efficiency, presented in a new Supplementary Table. While the results highlight that the high computational load required by their software may be a substantial burden for some analyses, they argue that their cost to speed from using individual level data is made up for by improvements to accuracy.

Author Response: We appreciate again for this thoughtful comment, and we agree that Supplementary Tables 14 and 15 (benchmarking the accuracy and computationally efficiency) can be informative for downstream users, informing the accuracy and runtime for various methods including both individual level-based and summary statistics-based methods. We have now added Supplementary Table 16 showing that parallel computing can increase the computational efficiency of the proposed method (mentioned in line 220-221), which can be also informative for downstream users.

3. More description of the models is now provided in the text, as well as a fuller description of the comparison benchmarking done.

Author Response: We thank the reviewer for this comment.

4. In response to my recommendation to benchmark additional unrelated phenotypes with differing genetic architectures, the authors provide some compelling figures in their Response to Reviewers. While not much detail is given about them, these results do appear to support the performance of their proposed model, however these additional phenotypes are not added to this manuscript. I am not sure I agree that the desire to save all other phenotypes to have a second applied publication can justify excluding any more here, as this article should serve as a proof of principle that their proposed method indeed can perform across the spectrum of polygenicity.

Author Response: We thank the reviewer for this comment. We agree with the reviewer that the results from additional phenotypes are compelling and do appear to support the performance of the proposed method. Therefore, we decided to include the results for a broader range of phenotypes such as basal metabolic rate, body fat percentage, whole body fat free mass, pulse rate, and educational attainment (instead of presenting them in a subsequent application paper) (shown in Figure R1 below and Figure 6 in the main text). We added a section in Results, titled as “Application to a broad range of complex traits”, with a detailed description (lines 280 – 315).

Figure R1. Estimated cross-ancestry genetic correlations for basal metabolic rate, body fat percentage, whole body fat free mass, pulse rate, and educational attainment

5. The authors have clarified their population label for the South Asian (formerly ‘Asian’) grouping. However, referring to the catch-all collection of all other ancestry individuals outside the tested primary labels as ‘admixed’ is not an appropriate terminology. Based on the description in the text, this does not seem to be a cohort comprising one population of admixed individuals but rather a collection of individuals from many different ancestry/population groups, some of whom are homogeneous. This grouping should be changed to ‘mixed’ throughout, as it is presented in their figures.

Author Response: According to this comment, the term, ‘admixed ancestry’, has been changed to ‘mixed ancestry’ (lines 87 and 569).

Reviewer #4

Most of the comments of reviewer 1 were well addressed by the authors.

Author Response: We thank the reviewer for this comment

1. Following up the third question of review 1, Is the proposed method robust to model misspecification? For example, the proposed method assumes an infinitesimal model where all the SNPs are causal SNPs. What if the true model is that only a small proportion of SNPs are causal? What if the causal SNPs are not 100% common for the populations.

Author Response: We thank the reviewer for this comment. We showed that the proposed method can provide unbiased estimates when only a small proportion of SNPs are causal (line 208-209 and Supplementary Table 12). For the case that the causal SNPs are not 100% common between the populations (or ancestries), we conducted a further

simulation mimicking a realistic scenario (also following a relevant suggestion by reviewer #5, Q2 below). We first simulated phenotypes using a set of causal SNPs ($MAF > 0.001$) that are 100% common between two populations. Then, we excluded SNPs of $MAF < 0.01$ and $MAF < 0.05$ for each population, which resulted in the scenario that the causal SNPs are not 100% common between two populations. The proportion of common SNPs are shown in Table R1.

Table R1: Percentage of the causal SNPs that are common between the two ancestries in the simulation when applying MAF QC differently

Ancestries	Percentage of causal SNPs common between two ancestries	
	MAF <0.01 (QC) applied for each ancestry	MAF <0.05 (QC) applied for each ancestry
White British vs other European	96.52	89.95
White British vs Asian	95.07	81.23
Other European vs Asian	94.44	81.28
Asian vs African	91.78	72.04
White British vs African	90.48	69.64
Other European vs African	89.80	69.65

In this scenario, we estimated genetic correlations using all SNPs ($MAF > 0.001$) and using common SNPs only ($MAF > 0.01$ or $MAF > 0.05$ for both ancestries) (Figure R2). The proposed method (using an optimal alpha) gives unbiased estimates when both simulation and estimation are based on all SNPs ($MAF > 0.001$). When using common SNPs only, the proposed method provides unbiased estimates unless the African ancestry group is analysed (Figure R2). When using common SNPs only, the existing method (fixing $\alpha = -0.5$) generates biased estimates except for White British vs. other European ancestry groups. In general, the proposed method is less biased compared to the existing method (using $\alpha = -0.5$). We have added these results (lines 209-212 and Supplementary Figure 4). We also discussed this as a limitation of the methods as “the causal SNPs may not be 100% common between ancestries and estimated cross-ancestry genetic correlations should be interpreted with caution although the proposed method produced less biased estimates compared to the existing methods” (line 395-398 in Discussion section).

Figure R2: Estimated cross-ancestry genetic correlations can be biased when the causal SNPs are not 100% common between ancestries, and the bias can be reduced by using the proposed method. The main bars are estimated cross-ancestry genetic correlations, and the error bars indicate 95% confidence interval (CI) of the estimates, empirically obtained from 100 replicates. The phenotypic simulation was based on the real genotypes, using ancestry-specific alphas (estimated from the real data as shown in Figure 1). The sample size used in this simulation was 5,000 for each ancestry (total 10,000). After QC including MAF < 0.001, the number of SNPs was ~200,000 across ancestry pairs, among which 10,000 SNPs were selected as causal. The true cross-ancestry genetic correlation was simulated as 1. For the genotypic data, QC of MAF < 0.01 or MAF < 0.05 was applied for each ancestry to generate a situation that causal SNPs are not 100% common between ancestries.

2. "It is noted that the computational efficiency of the proposed method can be increased further with parallel computing." Would the authors please elaborate more on how does the parallel computing work for this method?

Author Response: Table R2 below shows that the computational efficiency increases with parallel computing for the estimations of genomic relationship matrix (GRM) and genetic parameters by GREML (the main computational processes). The parallel function is easy to use with a single command in MTG2 software (-thread n, which can parallel the computation into n CPUs). This result is now in Supplementary Table 16 (mentioned in line 220-221).

Table R2: Computational efficiency of the proposed method with parallel computing

Conditions	Without -thread	-thread 10	-thread 20	-thread 50
	For GRM			
(10000+10000) individual and ~200k common SNP	Reading plink file: 3 min 5 sec Estimate GRM: 38 min 29 sec Record GRM: 16 min 41	Reading plink file: 3 min 3 sec Estimate GRM: 3 min 46 sec Record GRM: 16 min 4 sec	Reading plink file: 3 min 1 sec Estimate GRM: 2 min 3 sec Record GRM: 15 min 58 sec	Reading plink file: 3 min Estimate GRM: 1 min 24 sec Record GRM: 15 min 55 sec
	For GREML			
	Reading file: 10 min 3 sec For estimation: 17 min 7 sec	Reading file: 10 min 1 sec For estimation: 3 min 15 sec	Reading file: 10 min 0 sec For estimation: 2 min 16	Reading file: 10 min 3 sec For estimation: 1 min 36 sec

For estimation of GRM and estimation of genetic parameters by GREML, the computational efficiency increases with parallel computing. RAM is not changed with parallel computing, therefore not shown here (see Supplementary Table 15). Note that reading and recording are not computationally paralleled in the current version.

3. I was a reader of the authors' previous work, <https://academic.oup.com/bioinformatics/article/28/19/2540/289604?login=true>. Would the authors please comment on the most improvements compared with other old method in the angle of modelling? (I know this method is used to estimate cross-population genetic correlation)

Author Response: We thank the reviewer for this question. In terms of algorithm to solve linear mixed model equations and GREML, there is no difference between the old and proposed methods. However, in modelling genetic variance and covariance, the proposed method can account for the latent relationship between allele frequency and per-allele-effect size that can be differently distributed across ancestries. Without considering such relationship, the estimated genetic variance and covariance can be biased, and the association between allele frequency and per-allele-effect size has been shown by many studies¹⁻⁴.

Although we did not consider in this study, this can be extended to analyse two traits in a single population, i.e., the relationship between allele frequency and per-allele-effect size is not the same across different complex traits even considering samples within the same ancestry. For this case, the conventional algorithm of GREML should be revised because the genetic covariance structure is not the same between two traits, which is a new problem for the existing linear mixed model equations. This is beyond the scope of this study, but we will work on this for a subsequent method paper.

Reviewer #5

The authors have addressed most comments, but there are still concerns regarding the proposed method and its use in realistic data settings. In general, the authors have not sufficiently shown that their method performs better than existing methods (via simulations).

Author Response: We thank the reviewer for acknowledging that we have addressed most comments and also for their criticism. We actually show that the proposed method performs better than existing methods in various simulations (Figure 3, Supplementary Tables 11, 12 and 14, and Supplementary Figures 3 and 4). Following the reviewer's comment, we carried out additional simulations, i.e., allowing for some causal variants that have $MAF > 0.01$ in one population and $MAF < 0.01$ in the other population, and then, applying methods as usual, using only variants with $MAF > 0.01$ in both populations (please see point-by-point responses below).

1. It would be helpful to the reader for the author to concisely state how their proposed method differs from the existing methods, without the need for the reader to read 2 pages in the Methods. It appears that one reason the proposed method has gains over other methods is because of its use of individual-level data estimates (e.g. $var(x_i)$ instead of its expectation), which would be expected. If there is no varying alpha and summary-level data estimates are used is the proposed method the same as GRM4? More is needed in the main text to explain the difference between previous methods and the proposed method. This question was touched on by both Reviewers 1 and 3, but not satisfactorily addressed.

Author Response: We thank the reviewer for this comment. We have clarified how the proposed method differs from the existing method (GRM4), i.e. if there is no varying alpha between ancestries and alpha is fixed as -0.5, the proposed method is the same as GRM4 (explained in line 141 – 142). We also have clarified that the existing methods (GRM1 – 4) are individual-level data-based methods that are known to provide more accurate estimates, compared to summary statistics-based methods^{3, 5, 6} (elaborated in line 119-121).

2. The following comment of Reviewer 1 (bottom page 2) lists an important issue that the authors have not sufficiently addressed in their response

“It seems the authors used genetic variants present in both populations in their simulations. However, in reality, genetic variants private to a population may also impact diseases or complex traits. It would benefit the manuscript if the authors could demonstrate whether their method is also unbiased when ancestry-specific variants (variants present in one population but entirely absent in the other population) is included in the simulations.” Although the focus is on variants that have $MAF > .01$ in each population, the suggested realistic scenario should be considered in simulations. i.e., in simulations, allow for some causal variants that have $MAF > .01$ in one population and $MAF < 0.01$ in the other population. Then, apply methods as usual, using only variants with $MAF > 0.01$ in both populations.

Author Response: According to this comment (also a similar comment by reviewer #4, Q1 above), we carried out a further simulation mimicking a realistic situation. We first simulated phenotypes using a set of causal SNPs ($>MAF 0.001$) that are 100% common between two populations. Then, we excluded SNPs of $MAF < 0.01$ for each population, which resulted in the situation that the causal SNPs are not 100% common between two populations. The proportion of common SNPs are shown in Table R1

Table R1: Percentage of the causal SNPs that are common between the two ancestries in the simulation when applying MAF QC differently

Ancestries	Percentage of causal SNPs common between two ancestries	
	MAF <0.01 (QC) applied for each ancestry	MAF <0.05 (QC) applied for each ancestry
White British vs other European	96.52	89.95
White British vs Asian	95.07	81.23
Other European vs Asian	94.44	81.28
Asian vs African	91.78	72.04
White British vs African	90.48	69.64
Other European vs African	89.80	69.65

In this situation, we estimated genetic correlations using all SNPs ($MAF > 0.001$) and using common SNPs only ($MAF > 0.01$ for both ancestries) (Figure R2). The proposed method (using an optimal alpha) gives unbiased estimates when both simulation and estimation are based on all SNPs ($MAF 0.001$). When using common SNPs only, the proposed method provides unbiased estimates unless the African ancestry group is involved (Figure R2). When using common SNPs only, the existing method (fixing $\alpha = -0.5$) generates biased estimates except for White British vs. other European ancestry groups. In general, the existing method is more biased, compared to the proposed method. We have added these results (line 209-212, Supplementary Figure 4), and acknowledged that estimated cross-ancestry genetic correlations can be biased when the causal SNPs are not 100% common between ancestries, and the biasedness can be reduced by using the proposed method (discussed in line 395-398 as a limitation of the method).

Figure R2: Estimated cross-ancestry genetic correlations can be biased when the causal SNPs are not 100% common between ancestries, and the bias can be reduced by using the proposed method. The main bars are estimated cross-ancestry genetic correlations, and the error bars indicate 95% confidence interval (CI) of the estimates, empirically obtained from 100 replicates. The phenotypic simulation was based on the real genotypes, using ancestry-specific alphas (estimated from the real data as shown in Figure 1). The sample size used in this simulation was 5,000 for each ancestry (total 10,000). After QC including $MAF < 0.001$, the number of SNPs was $\sim 200,000$ across ancestry pairs, among which 10,000 SNPs were selected as causal. The true cross-ancestry genetic correlation was simulated as 1. For the genotypic data, QC of $MAF < 0.01$ or $MAF < 0.05$ was applied for each ancestry to generate a situation that causal SNPs are not 100% common between ancestries.

3. The current simulation setting is tailored to the proposed method and robustness needs to be demonstrated. When the competing methods are used in simulation settings tailored to themselves, GRM3 and GRM4 perform well in each ancestry combination (e.g., $\alpha = -0.5$ in both simulation and estimation, Fig 2). However, in the simulation setting of Fig 3, ancestry-specific alpha values (more realistic) are used with these methods having $\alpha = -0.5$ and the proposed method having alpha matching the simulation settings, which is an unfair comparison.

Author Response: The alpha parameter is a proxy of the relationship between allele frequency and per-allele effect size (i.e. heritability model^{3, 7}) as described in line 327-333. For real data analyses, existing methods (GRM3 and 4) assume that the true heritability model (as $\alpha = -0.5$) is invariant across ancestries. However, as shown in Figure 1, the estimated alpha values are not always -0.5 , indicating that the heritability model underlying the real phenotypes is different across ancestries. The key and novel feature of the proposed method is that it accounts for ancestry-specific alpha that is different across ancestries. However, existing methods do necessarily require the assumption of a homogeneous heritability model with $\alpha = -0.5$. Therefore, we attempt to overcome this limitation of the existing methods by proposing the method that can vary alpha. Based on the comment #1 from this reviewer, we clarified that if there is no varying alpha between ancestries and alpha is fixed as -0.5 , the proposed method is the same as GRM4 (please see line 141-142).

a. For the Fig 3 setting, it would be informative to run simulations with varying alpha and to have different (from the simulation settings) varying alpha in the proposed method.

Author Response: We appreciate the reviewer for this comment. The proposed method using alpha values that are far different from the true model, it would give biased estimates. Indeed, in Figure 3, GRM4 generates biased estimates because it uses different alpha values from the true model. It is noted that the proposed method is the same as GRM4 if alpha is fixed as -0.5 (Figure 2).

Nevertheless, we carried out additional simulations based on the reviewer’s suggestion (Figure R3). Figure R3 shows that the proposed method with optimal alpha values would provide unbiased estimates of cross-ancestry genetic correlations, confirming the results in Figure 3. However, when using mis-specified alpha that is different from the optimal alpha, the proposed method could provide biased estimates, which agrees with Figure 3 that incorrect alpha value generates biased estimates. When fixing alpha=-0.5 for both ancestries, the bias can be larger (GRM4 in Figure 3c).

The accuracy of estimated alpha based on model comparisons using the maximum log-likelihood has been verified by previous studies³, and we used the approach (AIC based on the maximum log-likelihood) to obtain an optimal alpha. We discussed a limitation as “we estimated optimal scale factors (α) with a moderate sample size especially for South Asian or African ancestry cohorts, resulting in a relatively flat curve of Δ AIC values. A further study is required to estimate a more reliable α for South Asian or African ancestry cohorts with a larger sample size” (line 394-395 in Discussion).

Figure R3: Estimated cross-ancestry genetic correlations can be biased when using a scale factor (alpha) that is different from the optimal alpha. The main bars are the estimated cross-ancestry genetic correlations, and the error bars indicate 95% confidence interval (CI) of the estimates, empirically obtained from 100 replicates. The sample size used in this simulation was 5,000 for each ancestry (10,000 in total), and the number of SNPs was 216,419, among which 10,000 SNPs were selected as causal variants. The true cross-ancestry genetic correlation was simulated as 1 (the horizontal dashed line).

b. For the Fig 2 setting, the proposed method should also be included (i) using alpha = -0.5 in the new method instead of varying alpha, to match the sim settings as for the other methods;

(ii) using varying alpha in the method, as proposed for the new method. **Author Response:** As noted in line 141, the proposed method using alpha = -0.5 for both ancestries is equivalent to GRM4. In fact, varying alpha values is the unique property of the proposed method, compared with the existing methods including GRM4. We thank the reviewer again for giving an opportunity to clarify the difference between the proposed and existing methods (lines 119-121 and 141-142), so that readers can better understand the property of the proposed method, compared to the existing method (GRM4).

These are the minimum simulation comparisons and as the authors are introducing a new method, a more extensive simulation study, with realistic settings, is needed.

Author Response: We conducted additional simulations with more realistic setting, as shown above (see the responses to Q2 and Q3). Some of these new results are now included in the manuscript to demonstrate that the proposed method can provide less biased estimates of cross-ancestry genetic correlations, compared to existing methods, in general (line 209-212, and 395-398, supplementary Table 13 and Supplementary Figure 4).

Reference

1. Evans, L.M., et al., *Comparison of methods that use whole genome data to estimate the heritability and genetic architecture of complex traits*. Nature genetics, 2018. **50**(5): p. 737-745.
2. Lee, S.H., et al., *Estimation of pleiotropy between complex diseases using single-nucleotide polymorphism-derived genomic relationships and restricted maximum likelihood*. Bioinformatics, 2012. **28**(19): p. 2540-2542.
3. Speed, D., et al., *Reevaluation of SNP heritability in complex human traits*. Nature Genetics, 2017. **49**(7): p. 986-992.
4. Schoech, A.P., et al., *Quantification of frequency-dependent genetic architectures in 25 UK Biobank traits reveals action of negative selection*. Nature communications, 2019. **10**(1): p. 1-10.
5. Ni, G., et al., *Estimation of genetic correlation via linkage disequilibrium score regression and genomic restricted maximum likelihood*. The American Journal of Human Genetics, 2018. **102**(6): p. 1185-1194.
6. Zhang, Q., et al., *Improved genetic prediction of complex traits from individual-level data or summary statistics*. Nature communications, 2021. **12**(1): p. 1-9.
7. Speed, D., et al., *Improved heritability estimation from genome-wide SNPs*. The American Journal of Human Genetics, 2012. **91**(6): p. 1011-1021.

REVIEWERS' COMMENTS

Reviewer #3 (Remarks to the Author):

In this manuscript by Momin et al., the authors propose a method for the generation of less biased cross-ancestry genetic correlations. They assess the performance of their method with multiple scaling factor options, and iron out a more appropriate scaling factor for anthropometric traits for several sets of empirical populations. Finally they test their method on simulated phenotypes as well several anthropometric phenotypes from the UK Biobank.

The authors had addressed the vast majority of my comments and concerns in the last round of review. In this second round, they further satisfied my primary concerns regarding more extensive benchmarking and comparison of their model to other existing models, as well as ensuring that their model performs well across a broader range of phenotypes. They also further investigated the important consideration of the impact of different causal SNPs across populations.

I appreciate the additional phenotypes included in the main text. One note regarding the new significant result for educational attainment – the authors note that “For educational attainment, there is significant genetic heterogeneity among African, south Asian, and European ancestries” (abstract line 34, as well as in other areas of the text) but do not discuss the non-genetic factors that will contribute immensely to this phenotype, e.g. socio-economic status, access to high quality public education, etc. It is well worth contextualizing this finding a bit to ensure that it is not misinterpreted.

Aside from this concern, I do not have any lingering major points, but caught a few small things:

- The term ‘other European’ is a bit confusing in context sometimes – for example line 287-288: “...it is significantly higher in South Asian than other European ancestry...”. This reads like South Asian is considered European – maybe consider capitalizing ‘other’ to make it clear this is one of your population definition – ‘Other European’.
- Line 84 – you updated the phenotypes included here now so should expand your list
- Line 87 – ‘a mixed ancestry group.’

Reviewer #4 (Remarks to the Author):

The comments were well addressed by the authors. I have no more questions.

Response Letter

Reviewer #3

In this manuscript by Momin et al., the authors propose a method for the generation of less biased cross-ancestry genetic correlations. They assess the performance of their method with multiple scaling factor options, and iron out a more appropriate scaling factor for anthropometric traits for several sets of empirical populations. Finally, they test their method on simulated phenotypes as well several anthropometric phenotypes from the UK Biobank.

The authors had addressed the vast majority of my comments and concerns in the last round of review. In this second round, they further satisfied my primary concerns regarding more extensive benchmarking and comparison of their model to other existing models, as well as ensuring that their model performs well across a broader range of phenotypes. They also further investigated the important consideration of the impact of different causal SNPs across populations.

Author Response: We are pleased that the reviewer is satisfied with our responses to their comments.

I appreciate the additional phenotypes included in the main text. One note regarding the new significant result for educational attainment – the authors note that “For educational attainment, there is significant genetic heterogeneity among African, south Asian, and European ancestries” (abstract line 34, as well as in other areas of the text) but do not discuss the non-genetic factors that will contribute immensely to this phenotype, e.g. socio-economic status, access to high quality public education, etc. It is well worth contextualizing this finding a bit to ensure that it is not misinterpreted.

Author Response: We thank the reviewer for this comment. We added the following in Discussion section. “For educational attainment, non-genetic factors, such as socio-economic status and access to public education, might also contribute to the phenotypes. In this study, we did not consider how the genetic effects interact with such environmental factors, i.e., genotype-by-environment interaction, which may partly underlie the significant genetic heterogeneity across ancestries. A further study is required to elucidate how genotype-by-environment interaction cause cross-ancestry genetic heterogeneity, especially for educational attainment”.

Aside from this concern, I do not have any lingering major points, but caught a few small things:

-The term ‘other European’ is a bit confusing in context sometimes – for example line 287-288: “...it is significantly higher in South Asian than other European ancestry...”. This reads like South Asian is considered European – maybe consider capitalizing ‘other’ to make it clear this is one of your population definition – ‘Other European’.

Author Response: We now use ‘Other European’ and “White British”.

-Line 84 – you updated the phenotypes included here now so should expand your list

Author Response: Expanded (line 86).

-Line 87 – ‘a mixed ancestry group.’
Author Response: Corrected (line 87).